# A method to construct the dynamic landscape of a bio-membrane with experiment and simulation

Albert A. Smith [1✉], Alexander Vogel [1], Oskar Engberg[1], Peter W. Hildebrand [1] & Daniel Huster [1]

Biomolecular function is based on a complex hierarchy of molecular motions. While biophysical methods can reveal details of specific motions, a concept for the comprehensive description of molecular dynamics over a wide range of correlation times has been unattainable. Here, we report an approach to construct the *dynamic landscape* of biomolecules, which describes the aggregate influence of multiple motions acting on various timescales and on multiple positions in the molecule. To this end, we use [13]C NMR relaxation and molecular dynamics simulation data for the characterization of fully hydrated palmitoyl-oleoyl-phosphatidylcholine bilayers. We combine dynamics detector methodology with a new frame analysis of motion that yields site-specific amplitudes of motion, separated both by type and timescale of motion. In this study, we show that this separation allows the detailed description of the dynamic landscape, which yields vast differences in motional amplitudes and correlation times depending on molecular position.

---

[1] Institute for Medical Physics and Biophysics, Leipzig University, Härtelstr. 16-18, 04107 Leipzig, Germany. ✉email: albert.smith-penzel@medizin.uni-leipzig.de

 1

Biomolecular function is determined, ultimately, by dynamics. The structure provides significant clues as to what a biomolecule might do, but ideally we want to see the molecule actually do it. Henzler–Wildman and Kern say that the "dream is to 'watch' proteins in action[1]," and point out that molecular dynamics (MD) simulation is unique in its ability to provide the time-resolved motion of atoms. Still, simulations are limited in sampling the conformational space and by inaccuracies in simulation parameters and thus require experimental validation; even then, it is not trivial to connect measured parameters to motion in a simulation. Considerable complexity arises due to different modes of motion: local librations, rotations around bonds, reorientations of molecular domains and the entire molecule, and collective motions of molecular assemblies contribute to dynamics[2]. Understanding biological systems requires determining which of these motions contribute to function, remembering that this contribution can be direct or indirect. Thus, to understand the function it is necessary to comprehensively describe the dynamics of a molecular system, requiring separation and parameterization of multiple contributions using all experimental and MD data available.

Nuclear magnetic resonance (NMR) is powerful because experiments provide site-specific motional information, where bond reorientation modulates interaction tensors (e.g., dipolar/quadrupolar couplings)[3,4]. Measurement of one-bond residual couplings provides an order parameter, $|S|$, defined as $|\delta_{resid.}/\delta_{rigid}|$, where $\delta_{resid.}$ is the averaged anisotropy of a coupling divided by the rigid limit of the coupling. Then, motions in a molecule leading to the reorientation of the tensor contribute to the reduction of $|S|$ from 1, albeit with a somewhat complex dependence on orientations sampled.

$|S|$ does not provide timescale resolution, whereas NMR relaxation rate constants are proportional to $(1 - S^2)$[5–8] and are selective for motions having correlation times $(\tau_c)$ matched to the eigenfrequencies $(\omega)$ of the spin system $(\omega\tau_c \approx 1)$; these frequencies can be varied by choice of the experiment (strictly speaking $|S|^2$ obtained from residual couplings may not exactly equal $S^2$, which determines relaxation behavior, unless an axis of symmetry for the motion exists). This timescale selectivity helps in separating motions, but for complex systems, a complete parameterization is rarely possible, and parameterization using simplified models often creates bias[9]. Alternatively, using an experimentally validated MD simulation[10], one should be able to extract and parameterize the specific motions. To attempt this, we consider a critical biological system: the lipid membrane, for which the complex dynamics of an extended molecular system is encoded into the reorientational motions of the single lipids.

The lipid membrane is nature's most important interface. It maintains a barrier function and provides the environment for various biological processes, i.e., communication and transport mediated by embedded proteins[11–13]. To enable these functions, lipid molecules are characterized by a highly dynamic structural polymorphism resulting in a well-balanced equilibrium of order and disorder[14]. This is best described by a *dynamic landscape* in which the crucial parameters are the correlation times of motion, their distribution widths, and the motional amplitude, where multiple motions yield a product of distributions. While spectroscopic tools and MD simulations have described individual aspects of this versatile dynamics[15–22], its comprehensive and quantitative description has not been presented. Here, we suggest an analytical method to use both NMR and MD data to quantitatively describe the dynamic landscape of a fully hydrated palmitoyl-oleoyl-phosphatidylcholine (POPC) bilayer. The method is based on dynamic detectors[9,23,24], which describe the timescale-specific generalized amplitude of motion of the C–H bonds of the POPC molecule.

In order to characterize the full dynamic landscape, we perform an extensive comparison of NMR/MD data using detector analysis, a method developed to eliminate bias while providing a quantitative, timescale-specific comparison of motions between different methods[9,23]. Next, we apply a frame analysis, which allows separating multiple types of motion using MD such that the product of time-correlation functions describing the individual motions yields the correlation function of the total motion[7,25–28]. For the separated motions, we explicitly fit distributions of correlation times for each motion, using only a few parameters. Combining the fits, we obtain a detailed characterization of the multidimensional dynamic landscape of a lipid membrane over several decades of correlation times. This methodology is well suited to quantitatively describe membrane dynamics in response to membrane protein function, lipid domain structure, and binding of molecules to the bilayer, and may be more generally extended to the characterization of other molecular systems.

## Results

**Experimental and simulated data analysis and comparison.** A series of seven NMR relaxation experiments ($^{13}$C $T_1$ at three fields, heteronuclear $^1$H–$^{13}$C nuclear Overhauser effect, and $^{13}$C $T_{1\rho}$ at three spin-lock strengths) and measurement of residual $^1$H–$^{13}$C dipole couplings (DIPSHIFT) were performed and analyzed using the dynamics "detectors" method[23], providing bond-specific dynamics information (Supplementary Note 1 contains further details of experimental data analysis). Detector analysis provides several detector responses, $\rho_n^{(\theta,S)}$, each of which characterizes motion within a specific window of correlation times ($\tau_c$). More precisely, $S^2$, the generalized order parameter, is a function of the orientations sampled by a given tensor, so that $1 - S^2$ typically increases with the amplitude of motion. Then, a detector response describes the fraction of $1 - S^2$ that results from motion for a given range of correlation times, i.e., the timescale-specific generalized amplitude of motion. We assume that the correlation function of reorientational motion is a decaying, multi-exponential function of the following form[7,26,29]:

$$C(t) = S^2 + (1 - S^2) \int_{-\infty}^{\infty} \theta(z) \exp(-t/(10^z \cdot 1s)) dz$$
$$0 \le 1 - S^2 \le 1, \quad \int_{-\infty}^{\infty} \theta(z) dz = 1. \tag{1}$$

$C(t)$ has an initial value of 1, and decays towards $S^2$, the generalized order parameter. $\theta(z)$ determines how different decay rate constants contribute to the total decay, where $z$ is the log-correlation time, $z = \log_{10}(\tau_c/s)$ (Eq. (1) is a Laplace transform). A detector response, $\rho_n^{(\theta,S)}$, then characterizes the function, $(1 - S^2)\theta(z)$, according to

$$\rho_n^{(\theta,S)} = (1 - S^2) \int_{-\infty}^{\infty} \theta(z) \rho_n(z) dz. \tag{2}$$

The amplitude of a detector response indicates the amplitude of motion for a specific range of correlation times, with that range defined by the detector sensitivity, $\rho_n(z)$. Detector responses are obtained via linear recombination of experimental data as previously defined[23,24]. The linear combinations for a set of detectors are chosen in order to extract the maximum information from the experimental data, i.e., obtain the best fit, and to obtain narrow, separated detector sensitivities.

If $(1 - S^2)\theta(z)$ is known, then detector responses, $\rho_n^{(\theta,S)}$, can be precisely determined by Eq. (2). Obtaining $(1 - S^2)\theta(z)$ from the $\rho_n^{(\theta,S)}$, however, is not possible without further assumptions;

therefore, interpretation must remain more loose; $\rho_n^{(\theta,S)}$ do not yield exact correlation times, and amplitude depends on correlation time, via $\rho_n(z)$. We can interpret detectors as windows into the total motion—selecting out only motion with correlation times near the center of the detector's sensitivity (the center is indicated on plots throughout). If $(1 - S^2)\theta(z)$ is smooth near a detector's center ($z_n^0 = \log_{10}(\tau_n^0/s)$), then the detector response is approximately equal to $(1 - S^2)\theta(z_n^0)$ multiplied by the detector width $(\Delta z_n)$[23]. For a distribution consisting of a few discrete correlation times, interpretation is different: a moderate detector response could result from a low-amplitude motion having a correlation time near the center of the detector, or from a high-amplitude motion with a correlation time away from the detector. Note that detectors can be viewed as an "ideal" relaxation rate constant; as motion increases in the sensitive range of a rate constant or detector, both parameters increase, but detector sensitivities are narrower than those for typical rate constants and are also normalized to one for easier interpretation.

While some ambiguity exists in detector interpretation, this is deliberate: by avoiding a specific model, we can quantitatively compare NMR to MD without introducing bias at the outset that could result from an incorrect model. To make this comparison, we calculate the reorientational correlation functions of H–C bonds from an 8.4 μs simulation of a POPC bilayer of 256 lipid molecules. This yields the correlation function in Eq. (1), and we may calculate detector responses from the MD simulation using a similar approach as is applied for NMR analysis[30]; details of both analyses are found in the "Methods" section.

The results are plotted in Fig. 1, where 6 detectors are obtained for 18 resolved NMR signals of POPC (Supplementary Fig. 1). Sensitivities of these detectors are shown in Fig. 1a, where $\rho_1$–$\rho_3$ cover ps/ns motion (~0.1–4 ns), and $\rho_4$–$\rho_5$ cover μs motion (6–70 μs). $\rho_0$ is sensitive to all motion falling outside the other windows. From MD, we could reproduce $\rho_0$–$\rho_3$, although the 8.4 μs trajectory is too short to compare motions with correlation times in the microsecond range. The agreement between NMR and MD in Fig. 1b is very good. The most significant outliers occur in the head group (α, β, γ), where MD underestimates $\rho_0$ and overestimates $\rho_1$, indicating that MD underestimates the rate of motion in the head group, most likely related to force field imperfections[10]. In the oleoyl and palmitoyl chains, we do not have a full site-specific resolution in NMR, but we do have excellent agreement with averaged detector responses obtained with MD (oleoyl C9/C10 are resolved in NMR). Then, we can combine results from MD in the chains (except oleoyl C9/C10) with experimental results elsewhere to illustrate the molecular motion of POPC in Fig. 2. In Fig. 2a, we map $\rho_0^{(\theta,S)}$ onto POPC. The sensitivity of $\rho_0$ is nonzero over multiple ranges of correlation times (Fig. 1a), but it is predominantly resulting from fast (<110 ps) motion, which can be verified from MD analysis (Supplementary Fig. 7). Motion is fastest at the ends of the chains and in the γ position of the head group. From $\rho_0$–$\rho_3$, one sees that chain motion and head group motion is predominantly fast ($\rho_0$/ $\rho_1$, ~110 ps or faster), whereas the glycerol backbone moves significantly slower ($\rho_2$/$\rho_3$). Motion is slowest for g2, where the highest detector response occurs for $\rho_3$ (~3.7 ns). Glycerol g1 experiences similar responses for $\rho_2$ and $\rho_3$ and g3 has the largest response for $\rho_2$ (~790 ps). To help interpret detector responses, Supplementary Movie 1 shows a POPC molecule from MD, where detector responses are plotted onto the molecule while sweeping through different timescales.

**Separation of total rotation into independent motions.** Previous studies have similarly obtained full site resolution of POPC dynamics by combining MD and NMR, but have been limited to

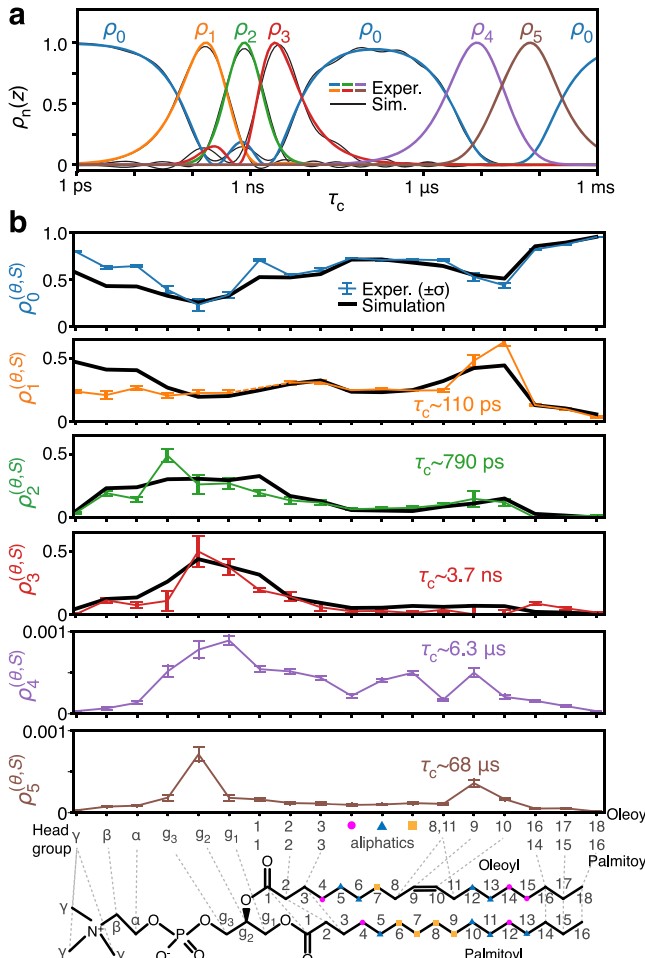

**Fig. 1 Experimental vs. MD-derived detector responses.** Panel **a** shows the sensitivity windows for six experimental detectors (color). We use MD data to approximate detectors 0–3, where the MD sensitivities are shown in gray. Detectors 4 and 5 are not calculated with MD. For detectors 1–5, the widths, $\Delta z_n$, in orders of magnitude, are 1.1, 0.8, 1.0, 1.3, and 1.3. In (**b**), the detector responses characterize the amplitude of motion in each window. Colored lines indicate the NMR detector responses and error bars indicate the 68% confidence interval, determined via linear propagation of error applied to the error of the experimental rate constants (see Supplementary Note 1, subsection 3 for details on experimental error determination). Black lines indicate the MD-determined amplitude, averaged over 256 copies of POPC in the MD simulation. Where experimental data are not resolved over multiple carbons, we average together the simulated data for the same carbons with uniform weighting for comparison to the experiment. Source data are provided as a Source Data File.

obtaining order parameters[19], or order parameters and the averaged effective correlation time[10]. In this study, we take another step forward and obtain timescale resolution, characterizing amplitude of motion for four ranges of correlation times. However, we would like to not only separate and characterize motion by timescale but also understand how different types of motions contribute to the overall dynamics. We note that the total rotation of an NMR interaction (e.g., a one-bond dipole coupling) can be decomposed into multiple rotations, as illustrated in Fig. 3, where rotation of an acyl H–C bond (parallel with the dipole coupling) in the lab frame is decomposed into four steps (Ω are the Euler angles of the rotation applied via the rotation matrix (**R**), resulting in a transformation between two

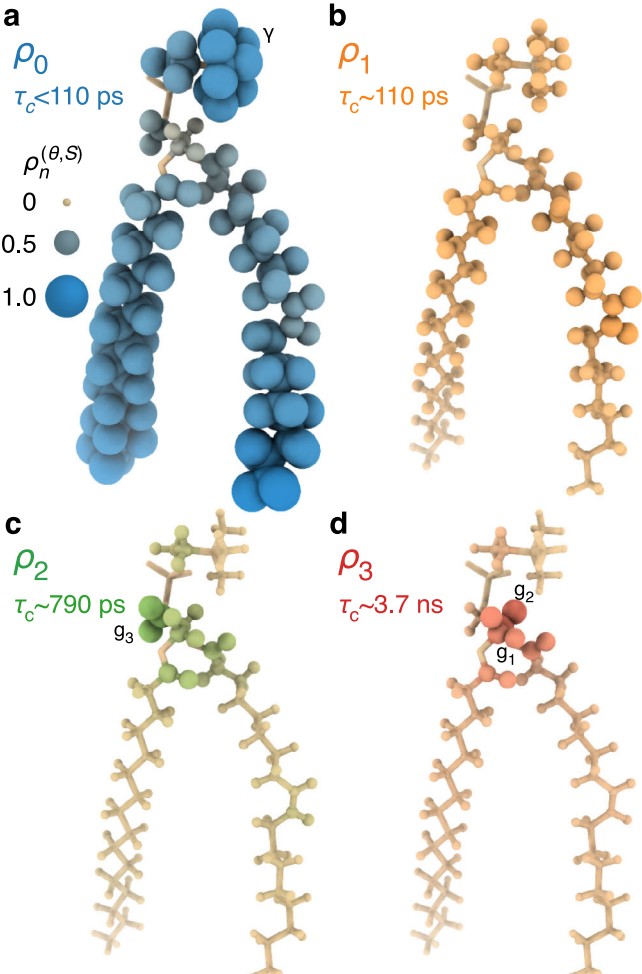

**Fig. 2 Detector responses mapped onto the individual segments of a POPC molecule.** Panels **a–d** show responses corresponding to $\rho_0$, $\rho_1$, $\rho_2$, and $\rho_3$, respectively. Each plot encodes the detector responses found in Fig. 1, where the radii of the H and C nuclei are proportional to $\rho_n^{(\theta,S)}$, and we fade from tan to color, depending linearly on $\rho_n^{(\theta,S)}$. The radius scale is indicated in the upper left of the figure. In cases where experimental data are ambiguous (some chain methylene segments), we plot MD-derived responses, noting that these responses are in very good agreement with the experiment (see Fig. 1). All 3D plots are produced with ChimeraX[67]. Source data are provided as a Source Data File.

individual frames):

$$\mathbf{R}(\Omega_t^{\text{total}}) = \mathbf{R}(\Omega_t^1) \cdot \mathbf{R}(\Omega_t^2(t)) \cdot \cdots \cdot \mathbf{R}(\Omega_t^n(t)). \quad (3)$$

Therefore, we apply a *frame* analysis, which allows the calculation of correlation functions for each motion[7,25–28]. Assuming statistically independent components, the correlation function of the total motion is given by the product of the correlation functions of the independent motions:

$$C(t) = C^1(t) \cdot C^2(t) \cdot \cdots \cdot C^n(t). \quad (4)$$

Separability of the correlation function is well established[7,25–27,31], but explicit separation from an MD trajectory has only been achieved for special cases[28], whereas we establish a general procedure here (note that if the individual correlation functions are multi-exponential, as in Eq. (1), then their product is also multi-exponential, see Supplementary Eq. 7).

Separation is achieved by defining a series of frames—a frame could be, e.g., the *z*-component of the moment of inertia (MOI)

for a carbon chain. Then, we can calculate a correlation function for the motion of a given H–C bond *within the frame* by rotating each frame of the trajectory such that the longest (*z*-) component of the MOI always lies on the *z*-axis, and evaluate the resulting H–C motion. Subsequently, we may determine how the motion *of the frame* results in the H–C bond motion, and calculate the corresponding function. The product of the resulting correlation functions is the total correlation function if: (1) terms resulting from motion within the frame are statistically independent of terms resulting from the motion of the frame, and (2) significant reorientation/reshaping of the residual NMR tensor brought about by motion within the frame occurs on a timescale significantly faster than the motion of the frame. Changes in tensor magnitude only are not subject to timescale separation. This procedure may be implemented iteratively to separate multiple correlation functions and is an explicit implementation of the theory developed by several groups including Brown[5,26], Brown et al. [6], Lipari and Szabo[7], Lipari et al. [32], Halle and Wennerström[25], Halle[31], and Wennerström et al. [33]. The extension of model-free theory from a theoretical principle to explicit implementation is detailed in the "Methods" section.

Using frame analysis, for each bond we separate three to four independent motions (Fig. 4a). To separate one-bond librations, we define a frame that aligns the C of the bond and all directly bonded atoms to a reference structure in order to capture local structural distortions. For the glycerol backbone ($g_1$, $g_2$, and $g_3$), head group (α, β, γ), and carbonyls (together abbreviated HG/BB), we define a frame that aligns the glycerol carbons and oxygens to separate overall motion from internal structural changes of the HG/BB. Similarly, in the chains (excluding carbonyls), we define a frame that aligns the longest (*z*-) components of the MOI (approximately the direction that chain points) to separate internal reorientation from the overall motion of the chain. Within the chains (excepting oleoyl carbons 9/10), we furthermore separate motion into components parallel and perpendicular to the MOI. Note that we do not explicitly introduce a local director frame, i.e., a frame that is perpendicular to the local membrane surface, because it is not well defined. Its motion will depend on the precise definition used, with the apparent dynamics depending on how many and which atoms are used to define the local membrane surface (see Supplementary Note 2 for precise definitions of each frame).

Results of the frame analysis (Fig. 4b–e) show varying behavior for the different motions. Responses from one-bond librations (Fig. 4b) are significantly smaller than all other motions and found exclusively in $\rho_0$ ($\tau_c < 110$ ps). Internal motions (Fig. 4c/d) are slower, with the largest responses for $\rho_0/\rho_1$, with the exception of the glycerol backbone, where motion is predominantly found in $\rho_2/\rho_3$ (790 ps/3.7 ns), being about 10× slower than other internal motion. Within the chains, motion perpendicular to the chain's MOI is slightly faster and typically has higher amplitude than motion parallel to the MOI. Overall motion (HG/BB or chain motion) differs considerably from internal motion: on average it is more broadly distributed (all $\rho_n^{(\theta,S)}$ have similar amplitudes), with significantly slower components, since $\rho_3^{(\theta,S)}$ (~3.7 ns) becomes large. Individual motions and detector responses can be viewed as movies in Supplementary Movies 2–6 (Supplementary Note 3 provides additional information about movies).

**Residual tensor evolution due to individual motions.** Detector analysis of individual motions captures contributions to $1 - S^2$ as a function of timescale. However, each motion samples bond orientations differently, which is not easily seen in Fig. 4. In

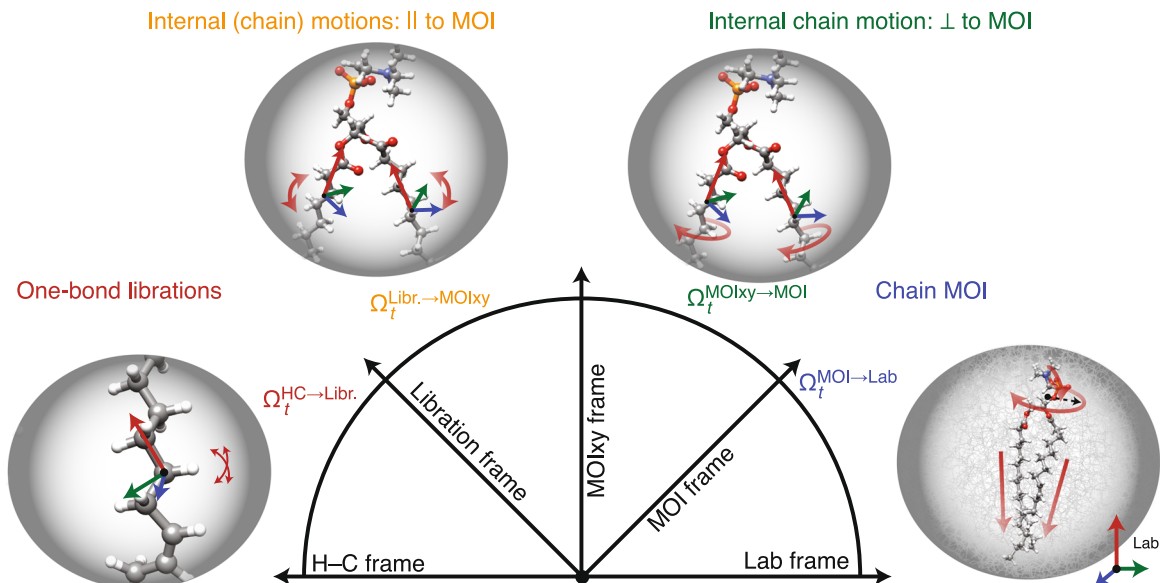

**Fig. 3 Illustration of the series of frames used for separating the motion of an H–C bond in an acyl chain in POPC into components (concept ref. [27]).** One-bond librations are separated by defining a frame that aligns the carbon of the H–C bond, and all atoms are directly bonded to that carbon. The motion of the bond within this frame defines local librational motion. Within each chain, librational frames move within the MOIxy frames. Alignment of the MOIxy frame removes rotation around the moment of inertia (MOI), so that this motion only contains motion parallel to the MOI. The motion of the MOIxy frames within the MOI frame then only contains motion perpendicular to the MOI. Finally, the chain MOI frames move within the lab frame, capturing the overall motions of the given acyl chain.

NMR, anisotropic interactions such as dipolar tensors are averaged by orientational sampling, resulting in a residual tensor of the interaction. This may be probed directly via the measurement of residual couplings to study the total molecular motion. However, it is not simple to separate individual motions' contributions to the residual tensor; spatial orientation of the residual tensor cannot be accessed with powder-averaged samples, and determining the sign also requires additional measurements such as the S-DROSS experiment[34,35]. Furthermore, the residual tensor from faster motion affects the relaxation induced by slower motion (see "Methods" for details). Therefore, in Fig. 5 we illustrate residual tensors resulting from each motion (estimated shape as $t \rightarrow \infty$), including time dependence for one carbon. The evolution of all tensors is shown in Supplementary Movies 7–11. Since librational motion has a very low amplitude, it results in minimal change to the residual tensor shape. Motion parallel to the chain MOI results in decreasing the longest component of the tensor and introduces anisotropy (Fig. 5b). This is expected since the parallel motion is a restricted rotation around one axis; sampling of a wider range of angles would result in a smaller tensor with larger anisotropy. Motion perpendicular to the MOI is a symmetric rotation around the MOI, resulting in significant reorientation of each tensor such that all residual tensors in chains align parallel to the MOI (Fig. 5c). The overall motion of the HG/BB causes residual tensors to nearly vanish, whereas MOI motion of the chains reduces tensor magnitudes by about 50% (Fig. 5d). Clearly, the frame analysis provides unprecedented insights into the molecular details of the motion of POPC in membranes and how motion relates to experimentally measurable parameters.

**Constructing the dynamic landscape.** In Figs. 1 and 2 we obtain a characterization of the distribution of motion, $(1 - S^2)\theta(z)$. However, attempting to estimate the specific form of $(1 - S^2)\theta(z)$ is unlikely to return completely reliable results, due to the non-uniqueness of the inverse-Laplace transform (ILT)[36], although one may nonetheless separate timescales with ILT[37]. In case we

know the specific form of $(1 - S^2)\theta(z)$ and it is described by only a few parameters, it might be possible to extract those parameters, but the presence of multiple motions, each described by different parameters, makes the problem intractable. On the other hand, once we separate the total motion into components, it is more reasonable to assume a simple functional form for the distribution of motion of each component. Then, detector responses for each bond and each motion are fitted to a three-parameter model of the distribution of motion, where parameters are correlation time, order parameter $(1 - S^2)$, and width. A skewed Gaussian distribution is used for internal motion, as might be expected for power-law behavior (collective motions[20,22,26,38,39], see Supplementary Note 4, subsection 1 for further discussion), and a regular Gaussian distribution for overall (HG/BB and chain MOI) motions.

The resulting MD-derived distributions may be found in Supplementary Fig. 14 and we plot the total distribution of motion in Supplementary Fig. 16, which results from the product of correlation functions (Eq. (4)). Although MD agrees well with the experiment (Fig. 1), we would like to perform a final refinement based on experimental results. To achieve this, we adjust the internal correlation time for each resonance. Where multiple positions in the POPC molecule overlap in the spectrum, we scale each correlation time by the same factor. Similarly, in the chains, we scale parallel and perpendicular components equally. Parameters from MD only and including experimental refinement are in Supplementary Fig. 13, and improvement in experimental agreement due to refinement is shown in Supplementary Fig. 17.

Experimentally refined distributions of motion are shown in Fig. 6a–d. While one cannot easily extract distributions from detectors describing overall motion, at the risk of losing important details of the individual motions, one may determine the total distribution by combining contributions from the individual motions; the result should satisfy the product in Eq. (4) (see "Methods"). The resulting dynamic landscape of the POPC membrane is shown in Fig. 6e.

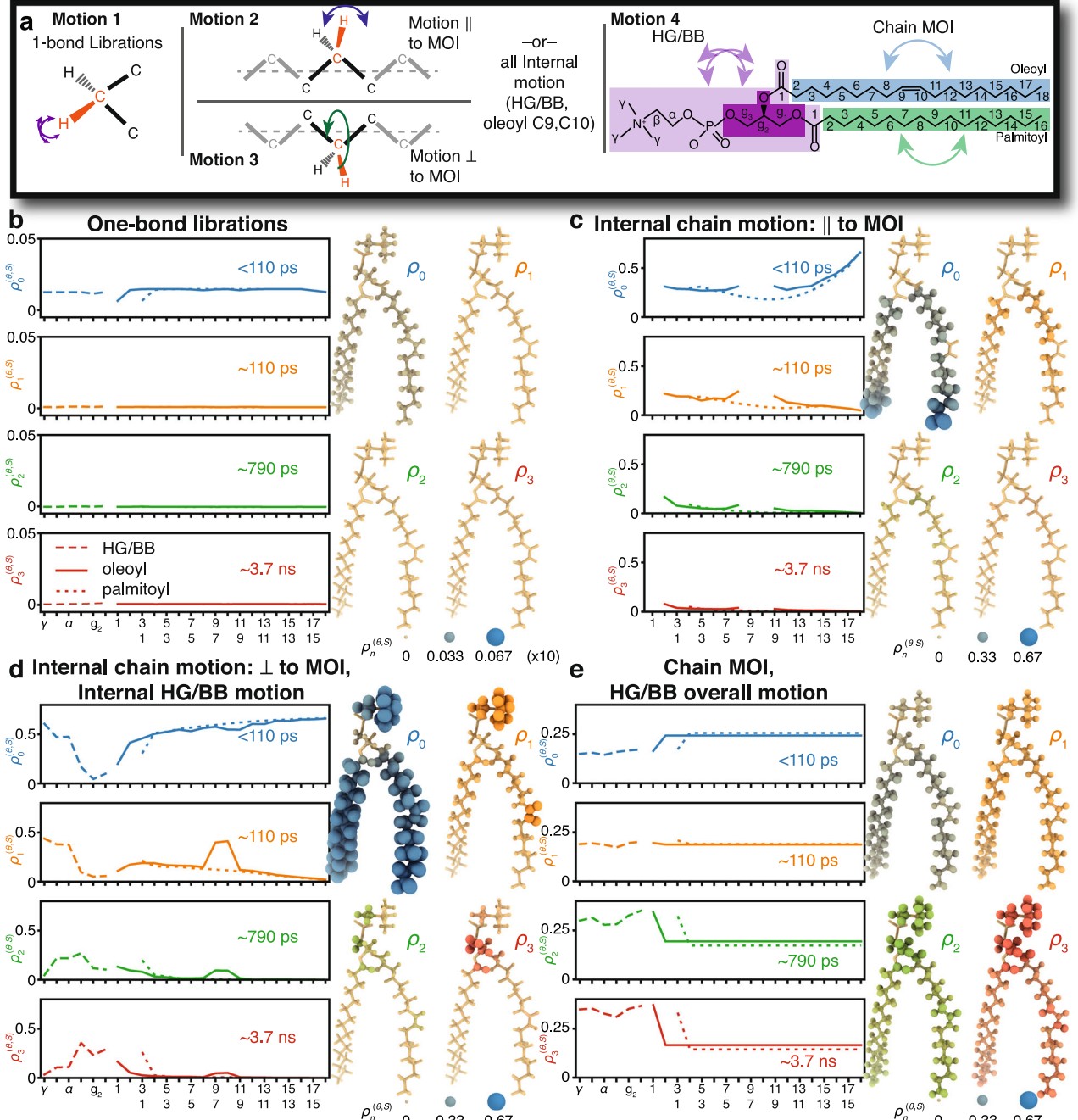

**Fig. 4 Detector analysis of individual frames.** Panel **a** illustrates the types of motion captured by the frame analysis: one-bond H–C (or C = O) librations are separated for all sites. Within chains, we separate motion parallel to the moment of inertia (MOI) and perpendicular to the MOI. For the head group, C′, and oleoyl carbons 9, 10 (double-bonded), we do not separate internal motion into two components. The overall motion of HG/BB is defined by the alignment of the glycerol atoms to a reference structure, and the overall motion of the chains is defined by the motion of the longest component (z-component) of each chain's MOI. **b**–**e** The detector analysis of the motions of each of the four motions, with detector responses also encoded onto the POPC molecule, where color intensity and radius depend on the detector response (same scale for (**c**–**e**), scale ×10 in (**b**)). For motion not split into perpendicular/parallel components, the internal motion (without libration) is shown in (**d**), with no data shown in (**c**). Source data are provided as a Source Data File.

## Discussion

Detector analysis provides a general method to quantitatively describe the dynamic landscape on which biomolecules exist based on experimental and/or simulated data. Detectors describe the timescale-specific generalized amplitude of motion, defined as the integral over the product of the distribution of correlation times of motion, $(1 - S^2)\theta(z)$, and the corresponding detector sensitivity, $\rho_n(z)$, as defined in Eq. (2). To assist in relating experimental

detector responses to the distribution of motion, we have color-coded the distributions in Fig. 6 according to the detector that is most sensitive to the corresponding correlation time. Then, if a sufficient number of detectors can be extracted from experimental and MD data, while separating motions with frames, the full dynamic landscape of a biomolecule can be constructed. Doing so, we rely on the detail provided by MD simulation, but refine results with the better accuracy of experiments.

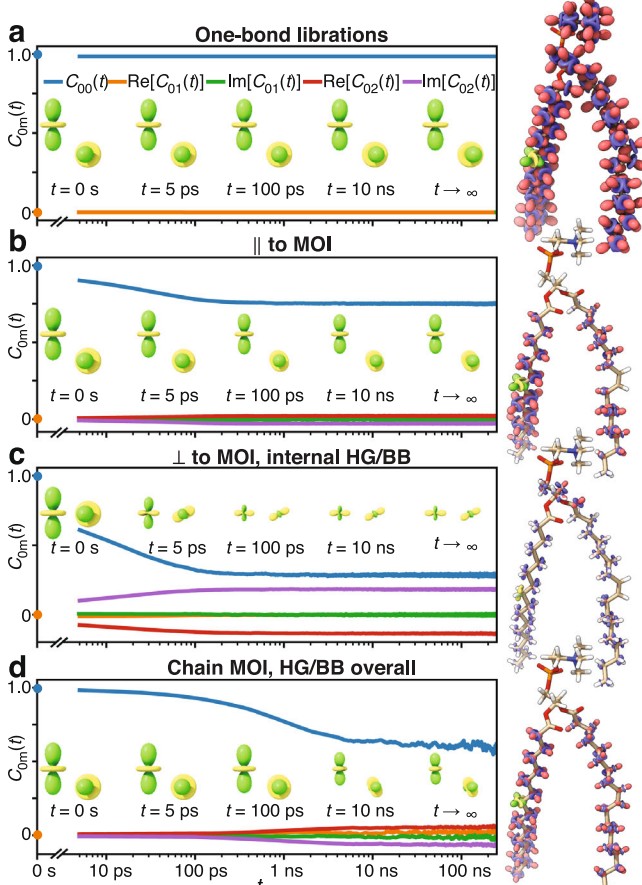

**Fig. 5 Tensor reorientation due to motion. a–d** The residual tensors resulting from individual motions are shown on the POPC molecule **a** Librational motion. **b** Motion parallel to the moment of inertia (no data for HG/BB and oleoyl carbons 9, 10). **c** Motion perpendicular to the MOI (chains), or all internal motion (one-bond librations removed) for HG/BB, and oleoyl carbons 9, 10. **d** Overall motions (chains defined by $z$-axis of MOI, HG/BB by RMS alignment of glycerol group). In **a–d**, pictures within the plots show the time dependence of the palmitoyl C8 tensor reorientation (green: positive; yellow: negative); the plots themselves show correlation functions defining the orientation of the C8 tensor ($C_{0p}(t) = \langle D_{0p}^2(\Omega_{\tau,t+\tau}^{\mathbf{v}:f-f})\rangle_\tau$, see "Methods"). In the molecule plots (right), red indicates the positive part of the tensor and blue the negative part. C8 (palmitoyl) is highlighted in the molecule plots in green and yellow. Source data are provided as a Source Data File.

The dynamics landscape (Fig. 6) brings the broad range of motion in the POPC membrane into stark relief, while still connecting that motion to the experimentally derived detector responses. The total amplitude of motion, $1 - S^2$, which can be obtained by integrating over correlation time, relates the molecular flexibility (the range of orientations sampled), whereas correlation time ($\tau_c$) reports the mobility (how quickly those orientations are sampled), both obtained as a function of molecular position[39]. In terms of the energetic landscape, amplitude informs us about the thermal equilibrium between states: a large value for ($1 - S^2$) implies that bonds undergo significant sampling of a large range of orientations, and these orientations must have similar free energy. Shorter correlation times indicate that the free energy cost of transitions between those orientations is low.

Consider the differences that can be seen in Fig. 6e: at the end of the chains and the head group, we have both high amplitude and short correlation times, so that almost all orientations are sampled, and the energetic cost of doing so is low (Supplementary Fig. 14). Contrast this to the backbone: amplitudes are slightly lower (similar to the middle of the palmitoyl chain, see $1 - S^2$ in Supplementary Fig. 13), but correlation times are much longer. Internal backbone motion (Fig. 6c) has a much higher free energy cost than internal chain motion. In Supplementary Note 5, subsection 1, we see that this cost comes from hops between several configurations of the backbone. This, however, is only a fraction of the total motion, the remainder coming from overall HG/BB motion (Fig. 6d), which is similarly slow. In this case, the energetic cost comes from the collectivity of motion[40]: concerted motions of many molecules in the membrane. This free energy is the sum of high enthalpy due to the large number of molecules involved and the entropic cost of the motion happening in a concerted fashion. Aside from long $\tau_c$, the collective motion also causes a broad distribution over correlation time. Collective motions do not happen over a fixed distance—we may have both short- and long-distance modes of motion—therefore we also observe a broad distribution of correlation times, with longer correlation times corresponding to longer distance motion. This leads to an essential property of the membrane: its elasticity[26,38,40–43]. While the backbone undergoes significant motion, a large part of that motion is collective, so that it does not result in the breaking of the local membrane structure. While one finds these modes via a combination of extensive field-cycling and temperature-dependent experiments[38,44], even then it is not possible to cleanly separate backbone collective motion from internal motion, since they occur on the same timescale; however, separation is achieved in Fig. 6c/d.

Distributions of correlation times are also observed for internal motions: in the backbone, this results from transitions between six configurations (Supplementary Note 5, subsection 1), and in the chains, numerous possible *trans/gauche* transitions broaden the correlation time distribution[45,46]. Collective dynamics also influence the chain MOI motion (Fig. 6d), although from Fig. 6e we expect the detector responses resulting from chain MOI to have a smaller impact on the total motion, due to masking by the large amplitude of the internal motion. Indeed, $\rho_2^{(\theta,S)}$ and $\rho_3^{(\theta,S)}$ in Fig. 1b show progressively lower responses. Although we cannot reliably predict $\rho_4^{(\theta,S)}$ or $\rho_5^{(\theta,S)}$ with only 8.4 μs of simulation, we would expect to see that collective motions (local director fluctuations) or diffusion around the curved liposome surface[47] are acting on small residual tensors from faster motions. The experiments agree, yielding the largest value for $\rho_4^{(\theta,S)}$ in the backbone where internal motion is smallest so that residual tensors are largest. In fact $\rho_5^{(\theta,S)}$ reaches its maximum at $g_2$ of the backbone, where internal motion is at its minimum. Note that the local maximum in $\rho_5^{(\theta,S)}$ at carbon 9 of the oleoyl chain is not so easily explained: from DIPSHIFT measurements, we know that the residual coupling is very small, so that collective motions acting on the residual coupling cannot explain the high detector response. A slow motion affecting primarily carbon 9 could be present, or alternatively, chemical exchange on a faster timescale could be mis-characterized as a slower reorientational motion.

Collectivity manifests in the distribution of motion, but we also find evidence of the different types of collectivity in the detector responses for each motion (Fig. 4). Nevzorov, Trouard, and Brown provide the spectral density as a function of the dimensionality, $d$, of the collective motion, according to a *power law*, yielding a spectral density, $J(\omega)$, which is then proportional to $\omega^{-(2-d/2)}$. In Supplementary Note 4, subsection 1, we show that this results from a distribution of correlation times having the form $\theta(z) \propto 10^{-z(1-d/2)}$. For a 2D collective motion, $\theta(z)$ is uniform, and detector responses vary proportionally to the detector

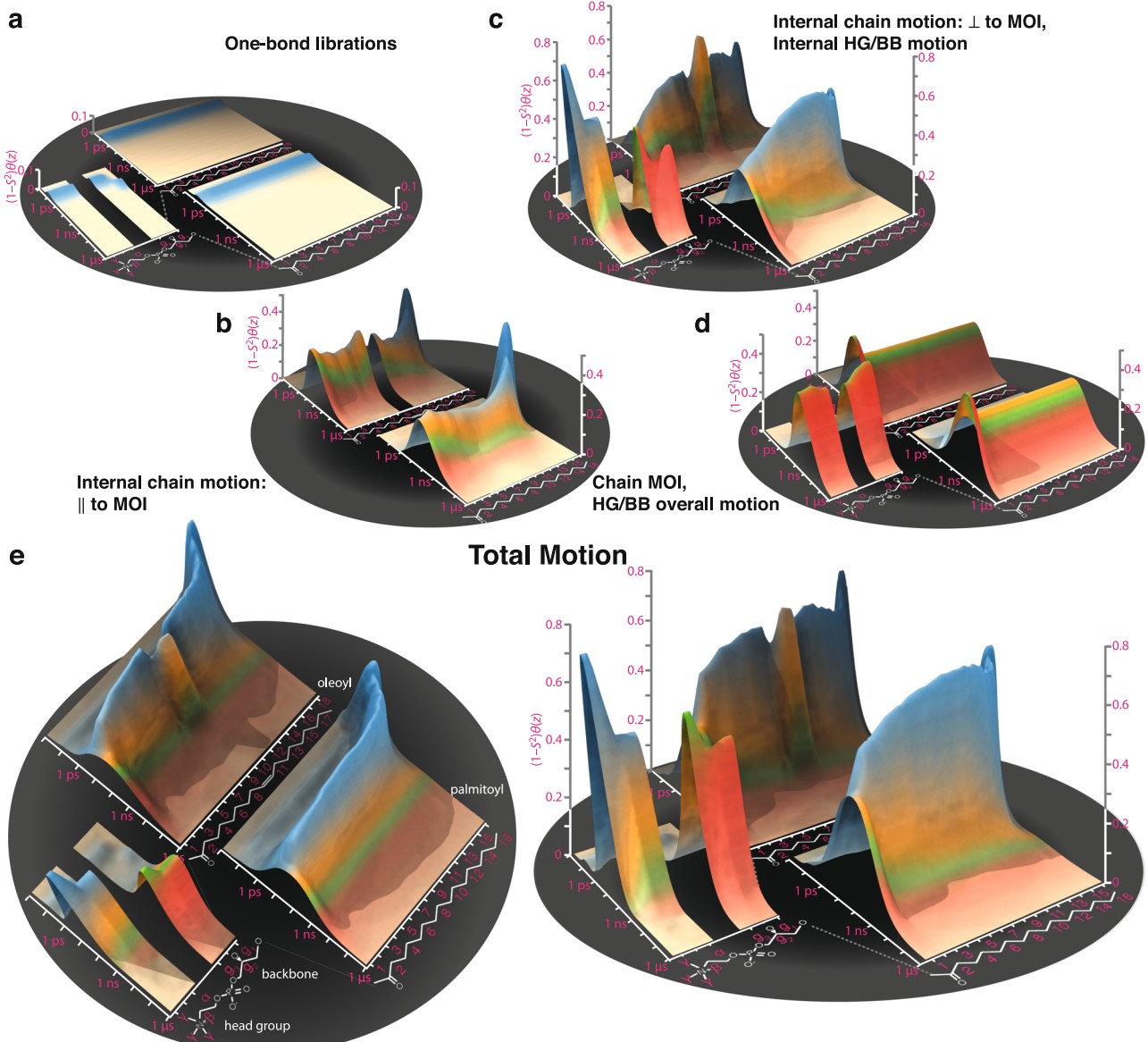

**Fig. 6 Dynamic landscape of POPC membranes. a–d** Fitted distributions of motion for the separated motion. Axes are the correlation time (left) and position (right), where each plot is broken up into parts (left: HG/BB; back: oleoyl; front: palmitoyl). The coloring corresponds to the experimental detector most sensitive to a given correlation time (blue: $\rho_0$; orange: $\rho_1$; green: $\rho_2$; red: $\rho_3$), where the intensity of the color is determined by the amplitude at the corresponding correlation time (fades to tan for small amplitudes). **e** The distribution of motion for the total motion, resulting from the product of correlation functions corresponding to the distributions in (**a–d**) (Eq. (4)). Details of fitting and calculating the total distribution are given in Supplementary Note 4. Source data are provided as a Source Data File.

width, $\Delta z_n$ (previously defined in ref. [23] widths are given in the caption of Fig. 1 and are relatively similar for all detectors). Indeed, for the overall motions of the HG/BB and chains (Fig. 4e), detector responses are approximately uniform. In contrast, faster internal motions are expected to behave according to a 3D power law. In this case, the detector responses also depend on the center position of the detector (given on a log scale as $z_n^0$ [23]). The resulting detector responses then scale according to $\rho_n^{(\theta,S)} \propto \Delta z_n 10^{-z_n^0/2}$, thus decreasing with increasing correlation time, consistent with the behavior in Fig. 4b, c.

Detector responses and the dynamic landscape predict several experimentally observable properties of membranes. For example, it has been shown that $S^2$ is proportional to $1/T_1$ [48,49]. $S$ describes the scaling of effective NMR tensors anisotropy from motion, where relaxation rates are proportional to the anisotropy of the tensors squared. Then, $T_1$ relaxation is induced by the scaled interactions, causing proportionality to $S^2$. However, this relationship only arises if $S^2$ is dominated by fast motion, and a relatively uniform, slower motion induces $T_1$ relaxation. The same relationship is found using detectors: $\rho_0^{(\theta,S)}$ is a good estimate of contributions to $1-S^2$ from fast motion, whereas $\rho_1^{(\theta,S)}$ captures the influence of slower motion on the relaxation. In Supplementary Note 5, subsection 2, we find that a linear relationship is maintained for $1-\rho_0^{(\theta,S)}$ and $\rho_1^{(\theta,S)}$ for detectors calculated both from experimental data and from the dynamic landscape in Fig. 6e.

The dynamic landscape also illustrates the origin of the $T_1$ temperature dependence in lipid chains: previous studies have not identified a $T_1$ minimum as a function of temperature for phospholipid chains in the liquid crystalline state [39]. One expects $T_1$ to reach a minimum when the maximum of the total

**Table 1 Experimental parameters.**

| Type | Acq. time (ms) | Scans | Reps. | Time pts | Second/last delay | $\omega_0/2\pi$ ($^{13}C$) (MHz) | $\omega_r/2\pi$ (kHz) | $\omega_1/2\pi$ |
|---|---|---|---|---|---|---|---|---|
| $\sigma_{HC}$ | 17.2 | 512 | 3 | 2 (on/off) | 10 s[a] | 150 | 5 | 0.1 W[b] |
| $R_1$ | 17.2 | 256 | 1 | 21 | 5 ms/4 s | 150 | 5 | 7.8 kHz |
| $R_1$ | 6.1 | 256 | 3 | 21 | 5 ms/4 s | 175 | 10 | 8.2 kHz |
| $R_1$ | 44.8 | 1024 | 4 | 21 | 5 ms/4 s | 100 | 5 | 5.3 kHz |
| $R_{1\rho}$ | 17.2 | 256 | 2 (4) | 18 | 0.2 ms/100 ms | 150 | 5 | 22.1 kHz |
| $R_{1\rho}$ | 17.2 | 256 | 2 (2) | 18 | 0.2 ms/100 ms | 150 | 5 | 12.0 kHz |
| $R_{1\rho}$ | 17.2 | 256 | 2 (2)[c] | 18 | 0.2 ms/100 ms | 150 | 5 | 7.0 kHz |
| DIPSHIFT | 17.2 | 512 | 5 | 33 | – | 150 | 2 | 65 kHz |

[a]Value is the length of the steady-state NOE delay (at least 3.4× $^{13}C$ $T_1$).
[b]Value not calibrated. $^1H$ saturation was established experimentally.
[c]Value within parentheses is the number of separate experiments performed with different offsets.

distribution of motion (Fig. 6e) approximately matches the center of $T_1$ sensitivity. From Fig. 6e, we are able to see why no minimum is observed: the maximum of the distribution falls at very short correlation times (1–10 ps), so that to have the $T_1$ sensitivity correspond with the maximum of the distribution would require significant cooling (causing the membrane to transition to the gel phase) or would require extremely high magnetic fields (~38 GHz $^{13}C$ Larmor frequency). Note that we would expect a $T_1$ minimum with respect to temperature for typical fields and temperatures in the glycerol backbone, where the distribution has a maximum of around 1 ns; perhaps, it is not a coincidence that Milburn and Jeffrey find a $T_1$ minimum in bilayers of egg phosphatidylcholine for the nearby $^{31}P$ atom just below ambient temperature[50].

Challenges remain in translating some of the concepts used to describe lipid dynamics into precise definitions that may be used for frame separation. Specifically, the local director was not used as a frame in this study, because it is not precisely defined. Still, rotational symmetry manifests in the alignment of tensors in Fig. 5c with their respective chains (and also leads to uniform behavior in Fig. 6d). Similarly, the complete separation of local and collective dynamics would require assigning a frame to each mode of motion, but collectivity still manifests as distributions of correlation times in the dynamic landscape.

On the basis of MD simulation and somewhat limited experimental data, detector analysis and the dynamic landscape provides broad insight into the dynamics of POPC membranes, and are consistent with the results of previous studies. Access to site-specific data for multiple types of motion, their correlation times, amplitude, and breadths based on experiment alone typically would require temperature dependence and field cycling, the latter of which prevents site-specific characterization without specific isotopic labeling[45,51]. Then, it is possible to validate the simulation with resolution in timescale using the detector analysis. Even so, it is not possible to fully parameterize multiple types of motion influencing H–C bond reorientation based on H–C correlation functions alone. Therefore, the motion was decomposed using a frame analysis, allowing us to analyze and fit motions separately, resulting in a fully parameterized dynamic landscape in Fig. 6. The landscape, via color coding, is still connected to the original detector windows used for experimental data analysis. This is a major strength of this method: we may always compare individual motions to total motions and experiment to simulation using the quantitative and timescale-selective detectors.

The dynamic landscape is critical to understanding motion in a complex system, where correlation time and amplitude are intimately related to the energy landscape. Capturing the form of the dynamic landscape, however, is only possible because we are able to perform the initial steps of analysis with limited assumptions

via detector analysis, thus avoiding biasing, and second by taking full advantage of the atom-specific information in MD to separate motion via frames, yielding simple distributions that may be reasonably parameterized. In lipid membranes, this information may be used to understand how the motion of the membrane couples to and allows or restricts motion in the proteins embedded within it. We expect future studies to investigate how membrane composition, including proteins themselves, influences these motions. However, we need not focus only on the membrane; our approach is general; protein and other molecular systems may be similarly measured and characterized. We may also modify our approach, using MD to identify locations and approximate timescales of critical motions and focus experimental efforts on those dynamics. Therefore, our approach may be the basis of a comprehensive understanding of motion and its function in complex systems.

## Methods

**Sample preparation.** POPC powder (Avanti Polar Lipids, Alabaster, AL, USA) was dissolved in 1:1 chloroform/MeOH and evaporated in a rotary evaporator at 40 °C. Afterwards, the sample was redissolved in cyclohexane and lyophilized overnight to acquire a fluffy powder and hydrated to 50 wt% using a HEPES buffer (10 mM HEPES, 100 mM NaCl, pH 7.4, prepared in Milli-Q $H_2O$). Multilamellar vesicles were produced by gentle centrifugation and ten freeze-thaw cycles between a 40 °C water bath and liquid nitrogen. Finally, the sample was inserted into a 3.2 or 4 mm NMR rotor.

**MD simulation.** A simulation of the membrane containing 256 POPC lipids was simulated for 8.37 μs. In addition, systems with 1024 and 4096 POPC lipids were simulated for 9.5 and 2.0 μs, respectively, to verify that trajectory size does not affect the dynamics characterized in this work (Supplementary Fig. 8). Each system was built in a rectangular periodic box and contained 42.1875 TIP3 waters[52] per lipid (50 wt%) and 0.1 M NaCl. Setup of the systems was conducted using published procedures[53–57]. Each system was energy minimized with the steepest descents algorithm and 1000 kJ mol$^{-1}$ nm$^{-1}$ as the threshold. All systems were equilibrated with harmonic positional restraints applied to lipids that were sequentially released in a series of equilibration steps. For each system, considerable time was spent on unbiased equilibration (500 ns) and the remaining trajectory was used for analysis. Coordinates were saved every 5 ps. The simulations were run in the NPT ensemble at a temperature of 298.15 K and a pressure of 1.0 bar using GROMACS 2019.2 and newer using the CHARMM36 force field[58]. Particle-mesh Ewald was used to treat electrostatic interactions, using a cut-off distance of 10 Å. Bonds involving hydrogen were constrained with LINCS[59] to allow a time step of 2 fs.

**Experimental setup.** Eight NMR experiments were acquired to characterize the dynamic behavior of POPC. All experiments were acquired as a series of 1D (pseudo-2D), $^{13}C$ detected spectra, with incrementation of a relaxation delay between 1D experiments. All experiments were initiated with a single pulse on $^{13}C$ (no polarization transfer from $^1H$), to avoid biasing of the measured dynamics due to dynamics effects on the polarization transfer. Pulse sequences are shown in Supplementary Fig. 2. Critical parameters for each experiment are given in Table 1. All experiments were acquired with at least 16 dummy scans for equilibration and SPINAL-64 decoupling[60]. Delays for relaxation experiments are approximately log spaced (first time point is always 0 s, second and final time point are listed). Separate $R_{1\rho}$ experiments with different offsets were acquired for different regions

of the spectrum, to minimize the variation of the effective field due to offset of the applied field[61]. For experiments with $\omega_1/2\pi$ of 12.0 and 22.1 kHz, two separate experiments were performed, so offsets were never >4.4 kHz, and for experiments with $\omega_1/2\pi$ of 7.0 kHz, four separate experiments were performed, so offsets were never >2.4 kHz. Field strengths for $R_{1\rho}$ experiments were verified via nutation experiments. The DIPSHIFT dephasing period used frequency-switched Lee–Goldburg decoupling for homonuclear decoupling[62], and SPINAL-64 for heteronuclear decoupling.

Relaxation rate constants are extracted from the series of 1D data by first fitting a reference spectrum using INFOS[63], and then fixing the positions and widths of the fitted peaks, but allowing amplitude and relaxation rate constants to vary (INFOS FitTrace function). All series were fitted to exponentially decaying functions ($\exp(-t \times R)$ or $1 - \exp(-t \times R)$, the latter for $T_1$ recovery). For DIPSHIFT, fitting was used to extract amplitudes from each 1D spectra, which were then separately fit in MATLAB, to explicit simulations of the DIPSHIFT sequence (simulation script provided as DIPSHIFT_sim.m in the MATLAB folder on https://github.com/alsinmr/POPC_frames_archive)[64].

**Detector analysis**. Both experimental data and MD-derived correlation functions are analyzed using the detector approach, described previously[23]. That is, experimental data is fitted with detector responses, minimizing

$$\min \sum_{\zeta} \sum_{n} \frac{(R_{\zeta}^{\exp \cdot} - [\mathbf{r}]_{\zeta,n} \rho_n^{(\theta,S)})^2}{\sigma(R_{\zeta})^2}, \quad (5)$$

where $\mathbf{r}$ is a matrix that has been optimized so that we obtain the set of detector sensitivities given in Fig. 1a. The $\mathbf{r}$ matrices used for experimental analysis can be found in Supplementary Tables 1–4. The resulting data fits are found in Supplementary Table 5.

We also use detectors to characterize dynamics from MD-derived correlation functions[26,31]. This differs from our and others' previous implementations, where correlation functions were first analyzed with an ILT[37], and Eq. (1) was computed explicitly[24]. The requirement for application of detector analysis is a linear relationship between the distribution of motion, $(1 - S^2)\theta(z)$, and the measured parameter, for an experimental relaxation rate constant, and a time point in a correlation function, these are

$$R_{\zeta}^{(\theta,S)} = (1 - S^2) \int_{-\infty}^{\infty} \theta(z) R_{\zeta}(z) dz$$
$$C(t) = S^2 + (1 - S^2) \int_{-\infty}^{\infty} \theta(z) \underbrace{\exp(-t/(10^z \cdot 1s))}_{R_{C(t)}(z)} dz. \quad (6)$$

We compare the relationship of an experimental relaxation rate constant to $(1 - S^2)\theta(z)$, to the relationship of a time point of the correlation function to $(1 - S^2)\theta(z)$. The two are essentially the same form, excepting the offset term, $S^2$. When fitting correlation functions, the term $S^2$ can be neglected: for a finite trajectory, it is not possible to differentiate the non-decaying fraction of the correlation function ($S^2$) from the very slowly decaying components. If we allow for correlation times at significantly longer than the trajectory, contributions from $S^2$ are simply absorbed into these long correlation times. Then, the sensitivity of a time point extracted from the trajectory is $R_{C(t)}(z) = \exp(-t/(10^z \cdot 1s))$. These sensitivities may be used to optimize detectors, using pyDIFRATE (provided via GitHub[64]), which implements the procedure previously described for experimental sensitivities (see Supplementary information of ref. [24]). Then, we optimize the MD sensitivities to match the first four experimentally derived detectors (Supplementary Note 1, subsection 5 provides further details).

**Frames analysis**. The total correlation function may be calculated as

$$C(t) = \langle D_{00}^2(\Omega_{\tau,t+\tau}^{\mathbf{v}}) \rangle_{\tau}$$
$$C(t_n) = \frac{1}{N-n} \sum_{m=0}^{N-n-1} \frac{3(\mathbf{v}(\tau_m) \cdot \mathbf{v}(\tau_{m+n}))^2 - 1}{2} \quad (7)$$

$D_{00}^2(\Omega_{\tau,t+\tau}^{\mathbf{v}})$ is the (0,0) component of the rank-2 Wigner rotation matrix element, operating on the Euler angles which rotate from the direction of an interaction tensor (collinear with the H–C or C=O bond) at some time $\tau$ to some later time $t + \tau$ (Euler angles given in the frame of the bond at the initial time, $\tau$). $D_{00}^2(\Omega_{\tau,t+\tau}^{\mathbf{v}})$ is averaged over all pairs of time points separated by $t$, indicated by the brackets, $\langle ... \rangle_{\tau}$, with averaging over the initial time, $\tau$. The latter equation gives a practical implementation of this formula, where $\mathbf{v}(\tau)$ are normalized vectors pointing in the direction of the bond ($D_{00}^2(\Omega)$ is equal to $(3\cos^2 \beta - 1)/2$, where the dot product of the normalized vectors yields $\cos \beta$). Equation (7) is used for calculating the correlation functions used for comparison to experimental analysis in Fig. 1.

In order to separate motions, we assume that we can define some *frame* for which a bond reorients due to reorientation of the frame. Then, the motion of the bond is the product of rotations *within the frame*, i.e., motion that is not correlated with the frame motion, and rotations *of the frame*. Suppose we have a bond at time $\tau$, expressed in its own frame (i.e., $\mathbf{v}^{\mathbf{v}}(\tau) = [0, 0, 1]$), and the same bond at some later time, given in the same frame, $\mathbf{v}^{\mathbf{v}}(t + \tau)$ (we use the superscript, v, to indicate the frame of the bond at the initial time). Then, the total rotation is given by

$$\mathbf{v}^{\mathbf{v}}(t + \tau) = \mathbf{R}_{ZYZ}(\Omega_{\tau,t+\tau}^{\mathbf{v}}) \cdot \underbrace{\mathbf{v}^{\mathbf{v}}(\tau)}_{=[0,0,1]} \quad (8)$$
$$= \mathbf{R}_{ZYZ}(\Omega_{\tau,t+\tau}^{\mathbf{v}:f}) \cdot \mathbf{R}_{ZYZ}(\Omega_{\tau,t+\tau}^{\mathbf{v}-f}) \cdot \mathbf{v}^{\mathbf{v}}(\tau).$$

We have simply broken the total rotation between the two vectors into two components: the first ($\Omega_{\tau,t+\tau}^{\mathbf{v}-f}$) is the rotation of the bond due to motion within the frame ($-f$ indicates that frame motion is removed), and the second ($\Omega_{\tau,t+\tau}^{\mathbf{v}:f}$) is the rotation of the bond due to motion of the frame (indicated by :f). Then, the same rotation occurring in Eq. (7) may be separated the same way, using the usual rules of spherical tensor rotations, as

$$C(t) = \langle D_{00}^2(\Omega_{\tau,t+\tau}^{\mathbf{v}}) \rangle_{\tau}$$
$$C(t) = \sum_{p=-2}^{2} \langle D_{p0}^2(\Omega_{\tau,t+\tau}^{\mathbf{v}:f}) D_{0p}^2(\Omega_{\tau,t+\tau}^{\mathbf{v}-f}) \rangle_{\tau} \quad (9)$$

At this stage, we first assume *statistical independence* of motion within the frame and motion of the frame[7,25], such that $\langle D_{p0}^2(\Omega_{\tau,t+\tau}^{\mathbf{v}:f}) D_{0p}^2(\Omega_{\tau,t+\tau}^{\mathbf{v}-f}) \rangle_{\tau} = \langle D_{p0}^2(\Omega_{\tau,t+\tau}^{\mathbf{v}:f}) \rangle_{\tau} \langle D_{0p}^2(\Omega_{\tau,t+\tau}^{\mathbf{v}-f}) \rangle_{\tau}$.

$$C(t) = \sum_{p=-2}^{2} \langle D_{p0}^2(\Omega_{\tau,t+\tau}^{\mathbf{v}:f}) \rangle_{\tau} \langle D_{0p}^2(\Omega_{\tau,t+\tau}^{\mathbf{v}-f}) \rangle_{\tau} \quad (10)$$

Second, we assume timescale separation, specifically, we require that there is some time, $t_1$, such that for $t < t_1$, $\langle D_{p0}^2(\Omega_{\tau,t+\tau}^{\mathbf{v}:f}) \rangle_{\tau} \approx \delta_p$, i.e., the orientation of the frame has not evolved significantly. For $t > t_1$, we require that the *shape*, although not necessarily the magnitude of the residual tensor due to motion within the frame stops evolving, such that $\langle D_{0p}^2(\Omega_{\tau,t+\tau}^{\mathbf{v}-f}) \rangle_t / \langle D_{00}^2(\Omega_{\tau,t+\tau}^{\mathbf{v}-f}) \rangle_{\tau} \approx \lim_{t \to \infty} [\langle D_{0p}^2(\Omega_{\tau,t+\tau}^{\mathbf{v}-f}) \rangle_t / \langle D_{00}^2(\Omega_{\tau,t+\tau}^{\mathbf{v}-f}) \rangle_{\tau}]$. The correlation function for these two limits becomes

$t < t_1$
$$C(t) = \sum_{p=-2}^{2} \delta_p \langle D_{0p}^2(\Omega_{\tau,t+\tau}^{\mathbf{v}-f}) \rangle_{\tau} = \langle D_{00}^2(\Omega_{\tau,t+\tau}^{\mathbf{v}-f}) \rangle_{\tau}$$
$t > t_1$
$$C(t) = \langle D_{00}^2(\Omega_{\tau,t+\tau}^{\mathbf{v}-f}) \rangle_{\tau} \sum_{p=-2}^{2} \langle D_{p0}^2(\Omega_{\tau,t+\tau}^{\mathbf{v}:f}) \rangle_{\tau} \lim_{t \to \infty} \frac{\langle D_{0p}^2(\Omega_{\tau,t+\tau}^{\mathbf{v}-f}) \rangle_{\tau}}{\langle D_{00}^2(\Omega_{\tau,t+\tau}^{\mathbf{v}-f}) \rangle_{\tau}} \quad (11)$$

We may then define two correlation functions, $C^{\mathbf{v}-f}(t)$, which describes motion within the frame, and $C^{\mathbf{v}:f}(t)$, which describes the motion of the frame, whose product yields $C(t)$ at all times, $t$.

$$C^{\mathbf{v}-f}(t) = \langle D_{00}^2(\Omega_{\tau,t+\tau}^{\mathbf{v}-f}) \rangle_{\tau}$$
$$C^{\mathbf{v}:f}(t) = \sum_{p=-2}^{2} \langle D_{p0}^2(\Omega_{\tau,t+\tau}^{\mathbf{v}:f}) \rangle_{\tau} \lim_{t \to \infty} \frac{\langle D_{0p}^2(\Omega_{\tau,t+\tau}^{\mathbf{v}-f}) \rangle_{\tau}}{\langle D_{00}^2(\Omega_{\tau,t+\tau}^{\mathbf{v}-f}) \rangle_{\tau}} \quad (12)$$
$$C(t) = C^{\mathbf{v}:f}(t) \cdot C^{\mathbf{v}-f}(t)$$

In this formulation, the terms $\lim_{t \to \infty} \langle D_{0p}^2(\Omega_{\tau,t+\tau}^{\mathbf{v}-f}) \rangle_{\tau}$ describe residual tensors resulting from all motion within the frame, similar to what is shown in Fig. 5. Normalization with $\lim_{t \to \infty} \langle D_{00}^2(\Omega_{\tau,t+\tau}^{\mathbf{v}-f}) \rangle_{\tau}$ shows us that it is only reshaping/reorientation of this tensor that must be timescale separated from reorientation of the frame itself, whereas the absolute magnitude may vary. That is, $\delta$, the anisotropy may continue to decay, but the variation of $\eta$, the asymmetry, and the Euler angles must remain timescale separated from frame reorientation.

Practically, $C^{\mathbf{v}-f}(t)$ is obtained by evaluating (7) in a time-dependent frame, which is aligned such that motion of the frame is removed. Take $\mathbf{v}_Z(\tau)$ to be a normalized vector giving the direction of the bond, and terms $\nu_\alpha^f(\tau)$ to be axes of the frame (we use $\nu_\alpha^f(\tau)$ as shorthand for each of the three vectors $\nu_X^f(\tau)$, $\nu_Y^f(\tau)$, $\nu_Z^f(\tau)$). We take $\Omega_{\tau}^f$ to be the set of Euler angles such that we obtain the $\nu_\alpha^f(\tau)$ by applying $\Omega_{\tau}^f$ to $X$, $Y$, or $Z$.

$$\mathbf{R}_{ZYZ}(\Omega_{\tau}^f) \cdot [1, 0, 0]' = \nu_X^f(\tau)$$
$$\mathbf{R}_{ZYZ}(\Omega_{\tau}^f) \cdot [0, 1, 0]' = \nu_Y^f(\tau) \quad (13)$$
$$\mathbf{R}_{ZYZ}(\Omega_{\tau}^f) \cdot [0, 0, 1]' = \nu_Z^f(\tau).$$

Then, we apply $\mathbf{R}_{ZYZ}^{-1}(\Omega_{\tau}^f)$ to $\mathbf{v}_Z(\tau)$ and subsequently calculate $\langle D_{00}^2(\Omega_{\tau,t+\tau}^{\mathbf{v}-f}) \rangle_{\tau}$:

$$\mathbf{R}_{ZYZ}^{-1}(\Omega_{\tau}^f) \cdot \mathbf{v}_Z(\tau) = \mathbf{v}_Z^{-f}(\tau)$$
$$C^{\mathbf{v}-f}(t_n) = -\frac{1}{2} + \frac{1}{N-n} \frac{3}{2} \sum_{m=0}^{N-n-1} (\mathbf{v}_Z^{-f}(\tau_m) \cdot \mathbf{v}_Z^{-f}(\tau_{m+n}))^2. \quad (14)$$

Calculation of $C^{\mathbf{v}:f}(t)$ is considerably more complex, requiring several terms depending on various elements of the Wigner rotation matrix elements. First, we require a consistent definition of the frame of the bond, given by time-dependent axes $\mathbf{v}_\alpha(\tau)$. As stated before, $\mathbf{v}_Z(\tau)$ should lie along the bond. Then, for an H–C bond, we take a C–C bond (containing the C from the H–C bond) to lie in the XZ-plane of the bond frame. From this, we can calculate $\mathbf{v}_X(\tau)$ and $\mathbf{v}_Y(\tau)$ (their definitions are arbitrary, as long as those definitions remain consistent, and the

axes are orthonormal). To obtain $C^{v;f}(t)$, we require the terms $\langle D^2_{0p}(\Omega^{v-f}_{\tau,t+\tau})\rangle_\tau$, describing the residual tensor of motion in the frame. These can be obtained by first finding the Euler angles $\Omega^{v-f}_\tau$ such that

$$
\begin{aligned}
\mathbf{R}_{ZYZ}(\Omega^{v-f}_\tau) \cdot [1,0,0]' &= \mathbf{v}^{-f}_X(\tau) \\
\mathbf{R}_{ZYZ}(\Omega^{v-f}_\tau) \cdot [0,1,0]' &= \mathbf{v}^{-f}_Y(\tau) \\
\mathbf{R}_{ZYZ}(\Omega^{v-f}_\tau) \cdot [0,0,1]' &= \mathbf{v}^{-f}_Z(\tau).
\end{aligned}
\tag{15}
$$

Using the resulting Euler angles, we then apply $\mathbf{R}^{-1}_{ZYZ}(\Omega^{v-f}_\tau)$ to the vectors $\mathbf{v}^{-f}_\alpha(t+\tau)$; the result is denoted as $\mathbf{v}^{v-f}_\alpha(t+\tau)$. These axes define the frame of the bond at time $t+\tau$, represented in the frame of the bond at time $\tau$, where the motion of frame $f$ is removed. Then, we finally define $\Omega^{v-f}_{\tau,t+\tau}$ to be the set of Euler angles that rotate to the bond from its frame at time $\tau$ to its orientation (in that frame), at time $t+\tau$. These may be inserted into the terms $\langle D^2_{0p}(\Omega^{v-f}_{\tau,t+\tau})\rangle_\tau$ required in (12). To obtain the limit as $t \to \infty$, we assume the MD orientational sampling is representative of the thermal equilibrium, and therefore pair all time points with all other time points, and take the average with equal weighting.

The rotation of the bond *within the frame* is calculated in the frame of the bond at its initial time. Then, the rotation of the bond due to the motion *of the frame* should be calculated in that same frame. To achieve this, we need the Euler angles rotating the bond from its initial orientation at time $\tau$, defined by $\mathbf{v}_\alpha(\tau)$, to its orientation at time $t+\tau$, but where the new orientation is only the result of the motion of the frame (motion in the frame removed). This is obtained by taking the $\mathbf{v}^{-f}_Z(\tau)$, where the motion of frame $f$ is removed at time $\tau$. Multiplication of each term by $\mathbf{R}_{ZYZ}(\Omega^f_{t+\tau})$ then yields the $\mathbf{v}^f_\alpha(t+\tau)$, which have been reoriented from time $\tau$ only by the motion of the frame. This can be seen below, where we see that the resulting terms are the result of the change in orientation due to the frame motion between times $\tau$ and $t+\tau$:

$$
\mathbf{v}^f_\alpha(t+\tau) = \mathbf{R}_{ZYZ}(\Omega^f_{t+\tau}) \cdot \mathbf{v}^{-f}_\alpha(\tau) = \mathbf{R}_{ZYZ}(\Omega^f_{t+\tau}) \cdot \underbrace{\mathbf{R}^{-1}_{ZYZ}(\Omega^f_\tau) \cdot \mathbf{v}_\alpha(\tau)}_{\mathbf{v}^{-f}_\alpha(\tau)}.
\tag{16}
$$

As before, we find the Euler angles defining the frame of the bond at time $\tau$, $\Omega^v_\tau$, and apply $\mathbf{R}^{-1}_{ZYZ}(\Omega^v_\tau)$ to the $\mathbf{v}^f_\alpha(t+\tau)$, yielding $\mathbf{v}^{v;f}_\alpha(t+\tau)$, which are the vectors at time $t+\tau$, in the frame of the bond at time $\tau$, where rotation between these times is due only to the motion of the frame. Then, we finally obtain the Euler angles yielding the $\mathbf{v}^{v;f}_\alpha(t+\tau)$, defined as $\Omega^{v;f}_{\tau,t+\tau}$, and insert these into the terms $\langle D^2_{p0}(\Omega^{v;f}_{\tau,t+\tau})\rangle_\tau$. The resulting terms may be used in Eq. (12), to obtain the correlation function due to the motion of the frame, $C^{v;f}(t)$.

This procedure may also be applied iteratively. Suppose we have two frames, denoted $f$ and $F$. Then, motion within frame $f$ is obtained as before, yielding $C^{v-f}(t)$. Motion due to frame $f$ is obtained by removing motion of frame $F$ first from both the bond vectors, $\mathbf{v}^{-F}_\alpha(\tau)$, and the vectors defining the axis of the frame $\nu^{f-F}_\alpha(\tau)$, and applying the procedure as described in the preceding paragraphs, yielding $C^{v;f-F}(t)$. Finally, motion due to frame $F$ is obtained by finding residual tensors within frame $F$ (we no longer need frame $f$ at this stage), and again applying the above procedure using the frame $F$. For each frame used, we require statistical independence of motion of the frame and motion in the frame and require timescale separation of residual tensor reorientation/reshaping due to motion in the frame and motion of the frame. Note that residual tensors in Fig. 5 are obtained for intermediate frames using the procedure described for obtaining the components $\langle D^2_{p0}(\Omega^{v;f}_{\tau,t+\tau})\rangle_\tau$, except that we use the terms $D^2_{0p}(\Omega)$ of the Wigner rotation matrix.

**Constructing a dynamic landscape.** Detector responses are defined by Eq. (1), and so for a given $(1-S^2)\theta(z)$, one may numerically integrate Eq. (1) to calculate $\rho^{(\theta,S)}_n$. Distributions may be fitted to the MD-derived values shown in Supplementary Fig. 12, which we achieved by performing a grid search over correlation time and distribution width, while optimizing the amplitude at every grid element, yielding a correlation function given by a discrete distribution:

$$
C^1(t) = S^2_1 + (1-S^2_1)\sum_i A_{1,i}\exp(-t/\tau_i)
\tag{17}
$$

Then, the product of two frames is

$$
\begin{aligned}
C^1(t) &= [S^2_1 + (1-S^2_1)\sum_i A_{1,i}\exp(-t/\tau_i)] \cdot [S^2_2 + (1-S^2_2)\sum_i A_{1,i}\exp(-t/\tau_i)] \\
&= S^2_1 S^2_2 + \sum_i [S^2_1 A_{2,i}\exp(-t/\tau_i) + S^2_2 A_{1,i}\exp(-t/\tau_i)] \\
&\quad + \sum_i\sum_j A_{1,i}A_{2,j}\exp(-t\frac{\tau_i\tau_j}{\tau_i+\tau_j}) \\
&= S^2_{12} + (1-S^2_{12})\sum_i A_{12,i}\exp(-t/\tau_i)\exp(-t/\tau_i).
\end{aligned}
\tag{18}
$$

The resulting amplitudes and correlation times can be numerically re-binned to obtain the new distribution, and the process is repeated to obtain the product of all motions as shown in Supplementary Fig. 16. Finally, we use NMR detector responses to refine the MD-derived result, by scaling the internal correlation time (or times) by a constant factor for all positions corresponding to each resolved resonance in the NMR spectrum, with results in Fig. 6.

**Reporting summary.** Further information on research design is available in the Nature Research Reporting Summary linked to this article.

## Data availability

NMR relaxation data generated in this study is tabulated in the Supplementary information and in the Source data file. Processed data found in the main text and Supplementary figures can be found in the Source data file. MD trajectories generated in this study have been deposited in the Zenodo database with 10 ns resolution at https://doi.org/10.5281/zenodo.5645031[65] and may also be viewed via MDsrv[66], using the following links: http://proteinformatics.org/mdsrv.html?load=file://public/papers/popc_dynamics/popc_256.ngl, http://proteinformatics.org/mdsrv.html?load=file://public/papers/popc_dynamics/popc_1024.ngl http://proteinformatics.org/mdsrv.html?load=file://public/papers/popc_dynamics/popc_4096.ngl. MD trajectories stored at 5 ps resolution are available upon request to the authors, where restricted access is due to the large size (several terabytes) of the trajectories. Source data are provided with this paper.

## Code availability

pyDIFRATE was used for the analyses presented here and is available as open-source software under the GNU General Public License. pyDIFRATE, as well as additional Python scripts and partial processing of MD trajectories, are provided to reproduce results (archived at Zenodo)[64]: https://doi.org/10.5281/zenodo.5642560.

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

## Acknowledgements

This study was funded by the Deutsche Forschungsgemeinschaft (DFG) through CRC 1423, project number 421152132, subproject A02 (to D.H., P.W.H.), through DFG Grant SM 576/1-1 (to A.A.S.), and through the Sigrid Juselius Foundation and Ruth and Nils-Erik Stenbäck's Foundation (to O.E.). Publication costs were supported by the Open Access Publishing fund of Leipzig University.

## Author contributions

A.A.S., A.V., and D.H. developed the concept of the frame analysis. A.A.S. developed the procedure and code for the frame analysis and performed NMR experiments and data analysis. A.V. and P.W.H. provided MD simulations. O.E. prepared samples of POPC. All authors contributed to the design of the study and manuscript preparation.

## Funding

## Competing interests

The authors declare no competing interests.
