## [Peer Review File · Nature Communications]

REVIEWER COMMENTS

Reviewer #1 (Remarks to the Author):

The authors propose an interesting new approach to constructing the energy landscape of a lipid membrane by combining molecular simulations with experimental NMR relaxation times as well as measurements of dipolar order parameters. The research is interesting and thought provoking and deals with an important biophysical problem. The figures are of high quality and illuminate the approach adopted. The research is an inherently mathematical work that is potentially suitable for Nature Communications upon revision.

From previous work it is known that the dynamics of lipid membranes encompass a broad distribution of correlation times. The paper generates various linear combinations of NMR relaxation rates and connects them to the correlation time distributions obtained from theoretical simulations. Because the correlation times span a broad range of logarithmic timescales, an effort is made to identify regions of the distribution that correspond to various possible lipid motions as detected by NMR relaxation. Essential to this construction is the assumption that the motions are both statistically independent and timescale separated. Stochastic modes of the motions are then identifiable with lipid structural features. The linear combinations corresponding to the various proposed lipid motions are then used to construct the energy landscape of the lipid membrane.

General comments:

1. I suggest that the authors include a general figure at the beginning of the paper to orient the general reader. E.g., SI Figure 11 might be added as a panel to Figure 1.
2. At the outset the general biophysical reader needs to be kept in mind. Please define clearly and explicitly all the terms used throughout the paper, which have different intuitive meanings for different general readers. These include "detector", "detector response", "sensitivity" etc. This should be done at the beginning of the paper to make the subsequent development clear both as mathematical definitions as well as regarding the physical meaning.
3. The connection of the mathematical analysis to the biophysics tends to be opaque. Every effort should be made to connect the mathematical analysis to experimental observables which may not just include NMR measurements.

4. The analysis should explicitly introduce or connect to the presence of a director frame for the collective lipid motions in the liquid-disordered phase. E.g., the time averaged coupling interactions in the director frame are connected to the appearance of the squared order parameter in the mathematical relaxation analysis.

5. The authors should comment on whether the detectors for a given component of the motion might lead to a relaxation extremum as temperature or frequency are varied. This is because previous studies (ref. 27) have shown that the relaxation times for all positions of the lipid molecules increase uniformly with temperature, and that relaxation time minima as temperature or frequency are varied are not seen. Are the authors able to relate the detector responses to this temperature variation?

6. Could the authors please comment on the conditions where the detector responses that underly the energy landscape might predict where a T1 minimum versus either frequency or temperature might be expected?

Specific comments:

Page 2, line 52 - The $(1-S^2)$ relation cannot just be stated a priori and it should be justified. The authors could cite: Brown, M. F., Seelig, J., and Häberlen, U. (1979) *J. Chem. Phys.* 70, 5045-5053, where this relation for a lipid membrane was evidently first presented. See also Brown, M. F. (1979) *J. Magn. Res.* 35, 203-215. In addition, it should be mentioned that S is the order parameter with respect to a preferred axis, i.e., the director axis of the lipid membrane.

Page 3, line 69 - please also cite: Brown, M. F., Ribeiro, A. A., and Williams, G. D. (1983) *Proc. Natl. Acad. Sci. USA* 80, 4325-4329, where many of these concepts were first discussed for lipid membranes based on NMR relaxation.

Page 4, line 109 - In the context of this relation please cite both: Lipari, G., and Szabo, A. (1982) *J. Am. Chem. Soc.* 104, 4546-4559 and Brown, M. F. (1982) *J. Chem. Phys.* 77, 1576-1599. The latter reference considers various possible formulations of the correlation time distribution, and the $(1-S^2)$ and S^2 dependence on order parameters is given by Eq. (3.19) and Eq. (4.9) of Brown (1982).

Page 7, line 156 - please also cite Brown (1982) here.

Page 7, lines 169-172 - Please also cite the generalized two-step approach of Brown (1982). The references cited are limiting cases for isotropic orientational averaging and are not directly applicable to lipid membranes. The results in Refs. 21-25 are obtained as the limits of the generalized two-step approach for isotopically reorienting systems (proteins, micelles). The results in Brown (1979); Brown, Seelig, and Häberlen (1979); and Brown (1982) should not be overlooked.

Page 10, line 246 - please consider citing Brown (1982) and Brown, Ribeiro, and Williams (1983) where are many of the concepts of collective lipid dynamics where first discussed in regard to NMR relaxation.

Page 11 Fig. 5 -for the total motion, how is the axial symmetry about the director axis manifested in the energy landscape?

Page 12, Discussion - at the outset please re-define "detector" for the general reader.

Page 13, line 304 - here it is necessary to cite: Brown (1982) and Nevzorov, A. A., and Brown, M. F. (1997) *J. Chem. Phys.* 107, 10288-10310. See also Chakraborty, S., et al. (2020) *Proc. Natl. Acad. Sci. U.S.A.* 117, 21896-21905.

Page 20, line 542 - please cite Brown (1982) here for the case of lipid membranes.

Supplementary information:

Page 2 - I would recommend also citing Leftin, A., et al. (2014) *Biophys. J.* 107, 2274–2286 for assignments.

Page 5 - it is necessary to clearly define or re-define "detector" and "sensitivity" as these terms have different meanings for different readers.

Page 6 - I would recommend explicitly defining "detector" in words and referring to the equation in which it is explicitly defined.

Page 18 - the authors should connect the coordinate frame analysis to the introduction of the director frame of the lipid bilayer. One possibility is to simply state that the director frame is taken as parallel to the lab frame thereby avoiding the orientation dependence. It is needed to clearly separate the time-dependent part of the interaction from the time independent part of the interaction. This is the most fundamental separation to be made, in my view.

Page 21-33 - please make a stronger effort to connect the results of the mathematical analysis to physical observables in the case of lipid membranes. The impact of the work will benefit from showing how the mathematical formulation leads to new biophysical insights.

Page 30, SI Figure 14 - the results for the overall motions seem to imply that a T1 minimum might be observed under some combination of frequency and temperature. Could the authors please comment on this possibility?

Page 32, SI Figure 16 - please comment on the cylindrical symmetry of the collective lipid motions about the director axis.

Michael F. Brown

Reviewer #2 (Remarks to the Author):

In the manuscript "A method to construct the dynamic landscape of a bio-membrane with experiment and simulation", the authors apply their recently developed "dynamics detectors" approach for analysis of conventional NMR relaxation and MD simulation data of fully hydrated bilayers of the previously thoroughly investigated phospholipid POPC, finally reporting their results as a "dynamic landscape" consistent with the extensive earlier literature on atomically resolved and multiple time scale studies of phospholipids. While all the individual aspects of the NMR experiments, MD simulations, "dynamics detectors" analysis, and "dynamic landscape" representation of the results can be found in the literature, the herein reported combined "method" can be taken as current state-of-the-art in investigations of molecular dynamics in lipid membranes and is of potentially great value for the broad community of biomembrane researchers. I recommend publication after only minor revisions as detailed below.

1) The actual novelty of the manuscript is diluted by the far too liberal use of terms such as “novel”, “we introduce”, “we develop”, “we obtain” for methodical aspects and molecular insights that are already well established in the literature. The text should be carefully edited throughout to give due credit to earlier literature and reserve the claims of novelty to the few – but significant – instances where it is warranted. More specifically:

a) Abstract, “We develop a dynamics detector method...”. The dynamics detectors method has already been described in a series of publications in *Angew Chem* and *J Chem Phys*: Refs 5, 15-17

b) Abstract, “...reveals vast differences in motion depending on molecular position.”. These differences are well known in the literature on NMR relaxation and MD simulation studies of lipid membranes and corresponding synthetic model systems with surfactants. Ref 18 is a very recent summary of this area.

c) Intro, “...we develop a novel frame analysis...” and “...we obtain a detailed characterization of the multidimensional dynamic landscape of a lipid membrane over several decades of correlation times”; page 9, “... we introduce a frame analysis...”; page 10, “... frame analysis provides unprecedented insights...”. Cite and explain novelty of the current frame analysis in relation to the local frame decomposition and detailed characterization in Lindahl 2001. *Molecular dynamics simulation of NMR relaxation rates and slow dynamics in lipid bilayers*. *J. Chem. Phys.* 115, 4938-4950. doi: 10.1063/1.1389469

d) Page 7. “...While obtaining full site resolution by combining MD and NMR is a step forward...”. Clarify what is actually the step forward in comparison to the summary in ref 18 and the extensive original literature obtaining full site resolution by combining MD and NMR.

2) In abstract, replace the too general term “lipid membranes” with the more specific “fully hydrated POPC bilayers” to more clearly connect to previous literature on this in detail investigated model system.

3) Intro. “...using an experimentally validated MD simulation, one should be able to extract and parameterize the specific motions.” Cite the recent summary on this subject in Ref 18 already here.

4) Intro. To the list of previous spectroscopy and MD studies in Refs 10-14, add these ones focusing on <10-100 ns time scale motion of relevance for the current manuscript:

Lindahl 2001. Molecular dynamics simulation of NMR relaxation rates and slow dynamics in lipid bilayers. *J. Chem. Phys.* 115, 4938-4950. doi: 10.1063/1.1389469

Ferreira 2015. Model-free estimation of the effective correlation time for C-H bond reorientation in amphiphilic bilayers: ^1H - ^{13}C solid-state NMR and MD simulations. *J. Chem. Phys.* 142, 044905. doi: 10.1063/1.4906274

5) Page 10. "... the sign [...] of the residual tensor cannot be accessed with powder averaged samples". The sign can actually be determined with the S-DROSS method even for powder samples. For the method cite Gross 1997. Dipolar recoupling in MAS NMR: A probe for segmental order in lipid bilayers. *J. Am. Chem. Soc.* 119, 796-802. doi: 10.1021/ja962951b

and for the specific case of S-DROSS applied to POPC cite Ferreira 2016. Acyl chain disorder and azelaoyl orientation in lipid membranes containing oxidized lipids. *Langmuir* 32, 6524-6533. doi: 10.1021/acs.langmuir.6b00788

6) Page 10. "...librational motion has very low amplitude...". Clarify that the librational motion is probably underestimated because of the LINCS constraint used in the simulation.

7) Page 10. "...attempting to estimate the specific form of $(1-S_2)\theta(z)$ is unlikely to return good results...". Explain that this is related to the non-uniqueness of the inverse Laplace transform, cite Istratov 999. Exponential analysis in physical phenomena. *Rev. Sci. Instrum.* 70, 1233-1257. doi: 10.1063/1.1149581

8) Page 10. Cite Lindahl 2001 for previous power-law fits of $C(t)$ from MD of lipid membranes.

9) Page 20. For inverse Laplace transform analysis of correlation functions, cite

Nowacka 2013. Signal intensities in ^1H - ^{13}C CP and INEPT MAS NMR of liquid crystals. *J. Magn. Reson.* 230, 165-175. doi: 10.1016/j.jmr.2013.02.016

10) Eqs 1-3. Clarify that the detector analysis relies on the assumption that the rotational correlation function $C(t)$ is given by a sum of exponentials (Eqs 1 and 2 in ref 5). Also mention that the relation between $C(t)$ and $\theta(z)$ can be written as a Laplace transform.

11) Eq 3. Explain how the product assumption in Eq 3 is consistent with the sum of exponentials assumption underlying the detector analysis in Eqs 1-3.

12) Figure 1. Give a tentative assignment of the microsecond dynamics captured by detectors 4 and 5 to some specific modes of motion (bilayer thickness fluctuations, lateral diffusion over curved bilayers,...). Also speculate over the underlying mechanism for the unexpected and previously not detected ~68 us mode of motion specific for carbon 9 at the double bond of the sn-2 acyl chain.

13) Figure 4. Move molecules to avoid overlap with figure axes.

14) Figure 5. Point out that similar "dynamic landscapes" could also be obtained by regularized Laplace inversion of $C(t)$ as Fig 5 in Nowacka 2013.

Reviewer #3 (Remarks to the Author):

In this work, the authors apply the concept of "dynamic detectors" to analyze motion in a lipid membrane. The idea of this analysis is to characterize the dynamics by constructing linear combinations of different NMR relaxation parameters that are expected to be specifically sensitive to different motional timescales. In this way, the detectors can in principle be used to identify the contribution of different time scales to the correlation function describing NMR relaxation. Although the authors claim it to be a "novel approach" in the abstract, it has been established for several applications in solid and liquid-state NMR. The novelty in this paper is therefore the application to the lipid membrane and the methods for separating out the contributions of different types of motion in the MD simulations to each detector (e.g. intramolecular vs overall motion etc.). Although the conclusions from that analysis are perhaps not unexpected, that may be because POPC lipid membranes are a very well-studied system, and application to other systems may prove more insightful.

Overall the paper is clearly written. Something that wasn't clear to me was how exactly the weights of the different observables in the detector functions are chosen. Is this done essentially by hand to locate a specific peak, or is there some systematic optimization done to target a desired profile/time window?

REVIEWER COMMENTS

Reviewer #1 (Remarks to the Author):

The authors propose an interesting new approach to constructing the energy landscape of a lipid membrane by combining molecular simulations with experimental NMR relaxation times as well as measurements of dipolar order parameters. The research is interesting and thought provoking and deals with an important biophysical problem. The figures are of high quality and illuminate the approach adopted. The research is an inherently mathematical work that is potentially suitable for Nature Communications upon revision.

From previous work it is known that the dynamics of lipid membranes encompass a broad distribution of correlation times. The paper generates various linear combinations of NMR relaxation rates and connects them to the correlation time distributions obtained from theoretical simulations. Because the correlation times span a broad range of logarithmic timescales, an effort is made to identify regions of the distribution that correspond to various possible lipid motions as detected by NMR relaxation. Essential to this construction is the assumption that the motions are both statistically independent and timescale separated. Stochastic modes of the motions are then identifiable with lipid structural features. The linear combinations corresponding to the various proposed lipid motions are then used to construct the energy landscape of the lipid membrane.

We thank the reviewer for his positive reception of our work, and for many useful suggestions especially to help better place our work in the context of existing concepts (esp. the director frame) and prior research.

General comments:

1. I suggest that the authors include a general figure at the beginning of the paper to orient the general reader. E.g., SI Figure 11 might be added as a panel to Figure 1.

Page 8, line 194

Thank you for this suggestion. We agree that some better orientation would be appropriate, and so have moved SI Figure 11 into the main text, although slightly further into the paper, as Figure 3, where the concept of the total rotation as a product of several individual rotations is introduced.

2. At the outset the general biophysical reader needs to be kept in mind. Please define clearly and explicitly all the terms used throughout the paper, which have different intuitive meanings for different general readers. These include "detector", "detector response", "sensitivity" etc. This should be done at the beginning of the paper to make the subsequent development clear both as mathematical definitions as well as regarding the physical meaning.

Page 4, line 101

This is a good point. While detector analysis is becoming familiar to us, We have added an extended discussion on the definition of terms and their interpretation on page 4-5.

3. The connection of the mathematical analysis to the biophysics tends to be opaque. Every effort should be made to connect the mathematical analysis to experimental observables which may not just include NMR measurements. We agree that comparison to experiment is crucial, and therefore have included the following changes.

Thank you for the suggestion. We've made several edits to better connect the mathematics to power laws and experimental observables.

Page 16, line 396

We discuss how detectors should behave for 2D and 3D power laws, noting that overall motions result in uniform detector responses, consistent with 2D power laws, and that internal motions (parallel/perpendicular) to chains have decreasing detector responses with increasing correlation time, consistent with 3D power laws

Page 17, line 414

We discuss how the $1/T_1$ vs. S^2 relationship manifests for detector analysis, finding similar behavior for ρ_1 and $1-\rho_0$, where our landscapes also return the linear relationship.

Page 17, line 425

We discuss that temperature dependence of T_1 relaxation (and the lack of T_1 minima) may be understood as a result of the distribution of correlation times having its maximum well outside the sensitive range of T_1 measurements at typical fields.

4. The analysis should explicitly introduce or connect to the presence of a director frame for the collective lipid motions in the liquid-disordered phase. E.g., the time averaged coupling interactions in the director frame are connected to the appearance of the squared order parameter in the mathematical relaxation analysis.

We agree that the lipid director frame plays an important role in the dynamics of membranes and in their analysis, however, it does not directly manifest in our computations, and so we discuss here why that is, and add some changes to the manuscript as well, to clarify this point.

In terms of analysis, either the bilayer director frame (following definitions from Brown 2019: the static frame that is normal to the membrane surface) or the local director frame (the dynamic frame, normal to the local membrane) provides a symmetry axis that allows simplification of the calculation of order parameters and correlation functions. However, as implemented in this study, these symmetry axes are not required. We have constructed our frame analysis to be more

general, and thus easily applied to systems such as proteins that may not have a convenient symmetry axis for the motion. Therefore, the lipid director is not discussed as a core part of the analysis method.

Nonetheless, one could argue that we should introduce the local director frame as one of the intermediate frames in the frame analysis in order to better capture collective motion. The challenge here is that while the concept of the local director frame is clear, its explicit definition is not. To define it, we had attempted a best-fit of a plane to one or more atoms of the lipid of interest and surrounding lipids (we used a Gaussian weighting to fit the plane, based on distance in the xy-plane from the lipid being analyzed). The problem is this: using a broader Gaussian removes the influence of shorter-range modes of motion, but using a narrower Gaussian then included local, likely non-collective motion of the individual lipids. Ideally, we do not want to introduce a frame that behaves differently based on variables in its definition, thus introducing somewhat arbitrary dynamics based on the details of its definition.

An interesting pathway forward would be to attempt to couple the frame analysis with a mode analysis (i.e. principal component analysis or Brüsweiler and Promper's iRED analysis), but this is beyond the scope of a first paper introducing the frame analysis. We should also mention that the manifestation of collective dynamics could be distorted by the length and size of the MD simulation, although in SI Fig. 8, we find that these distortions due to simulation size do not manifest until ~75 ns or so, and then only where the total amplitude of motion is already very low.

It is encouraging to note the emergence of distributions of correlation times, especially in Fig. 6D, where overall motion of the chains and the HG/BB are characterized. This broad distribution is the result of the local director motion coupling to collective modes of motion, and it arises without explicit usage of the director frame. The alignment of tensors within the MOI frames of the chains (Fig. 5C) is the result of the symmetry of motion around the MOI, which also results from the MOI vector's symmetric sampling of orientations about the director. Thus, even without its explicit inclusion, the importance of the director frame is not lost in our analysis.

Page 9, line 236

We state that the director frame cannot be explicitly defined.

Page 17, line 438

We discuss some of the remaining challenges in this type of analysis, specifically that the director frame cannot be extracted without precisely defining how many lipids define the local plane, and also that separating collectivity would require a frame for every mode. However, we note that the existence of symmetry about the director frame still leads to alignment of the residual tensors with the chains and uniform behavior in Fig 6D.

Page 17, line 414

The connection between the scaled order parameter and T_1 is now discussed, and we demonstrate a similar relationship for detectors, now shown in SI Fig. 19.

5. The authors should comment on whether the detectors for a given component of the motion might lead to a relaxation extremum as temperature or frequency are varied. This is because previous studies (ref. 27) have shown that the relaxation times for all positions of the lipid molecules increase uniformly with temperature, and that relaxation time minima as temperature or frequency are varied are not seen. Are the authors able to relate the detector responses to this temperature variation?

We agree with the observations made in ref. 27 that no minima are observed with magnets that are currently available. Because temperature variation and the emergence of T_1 minima is more easily examined using the dynamic landscape, as opposed to the detectors themselves, we note that T_1 minima occur when the maximum sensitivity of the T_1 experiment overlaps with the maximum of the distribution of correlation times. We can see fairly clearly that for most positions, this overlap would only occur for extremely high fields, with exceptions in the glycerol backbone.

Page 17, line 425

We discuss the occurrence of temperature minima (or lack thereof).

A related point is that our distributions are constructed so that they do have maxima in the distribution. This diverges slightly from the established power laws from Nevzorov, Trouard, and Brown (1997), for which the distributions corresponding to the power laws have, in principle, infinite integrals. It seems fairly clear that these formulae are not intended to describe the distribution extending over all correlation times, and would of course be valid within the range of correlation times to which they have been previously applied. For our purposes, however, it is important that the distributions have finite integrals. We have therefore also added a comparison of the selected distribution (skewed Gaussian) to the power-law based distribution to the SI (SI Fig. 11) finding that for $d=3$, the distributions are highly similar in the valid range of the power-law (exactly how one should terminate the power-law distributions remains an open question, but we believe that our choice is reasonable).

SI page 27, line 586

(referenced in main text page 12, line 308)

6. Could the authors please comment on the conditions where the detector responses that underly the energy landscape might predict where a T_1 minimum versus either frequency or temperature might be expected?

See response to previous point

Note that T_1 vs. frequency should never have a minimum for dipolar/quadrupolar induced relaxation, since the spectral density decreases monotonically for increasing frequency. Detectors, on the other hand, can have maxima, due to their normalization to 1, and we do observe these for some positions in Fig. 1.

Specific comments:

Page 2, line 52 - The $(1-S^2)$ relation cannot just be stated a priori and it should be justified. The authors could cite: Brown, M. F., Seelig, J., and Häberlen, U. (1979) J. Chem. Phys. 70, 5045-5053, where this relation for a lipid membrane was evidently first presented. See also Brown, M. F. (1979) J. Magn. Res. 35, 203-215. In addition, it should be mentioned that S is the order parameter with respect to a preferred axis, i.e., the director axis of the lipid membrane.

Page 2, line 56

We have added the requested citations

However, it is important to note that these citations require calculating S with respect to the director axis, but in fact, obtaining S^2 (defined as limit $C(t)$ as t approaches infinity) does not require such a reference frame. Szabo and co-workers treat this more generally, and so we also include appropriate citations to this work as well.

Page 2, line 58

Then, since the preferred axis is not required, we state more generally that $|S|^2$ from residual couplings is not exactly equal to S^2 responsible for relaxation behavior in the absence of an axis of symmetry (equality of these two parameters arises from presence of the director, or from another symmetry axis).

Page 3, line 69 - please also cite: Brown, M. F., Ribeiro, A. A., and Williams, G. D. (1983) Proc. Natl. Acad. Sci. USA 80, 4325-4329, where many of these concepts were first discussed for lipid membranes based on NMR relaxation.

Page 3, line 75

We have added the citation.

Page 4, line 109 - In the context of this relation please cite both: Lipari, G., and Szabo, A. (1982) J. Am. Chem. Soc. 104, 4546-4559 and Brown, M. F. (1982) J. Chem. Phys. 77, 1576-1599. The latter reference considers various possible formulations of the correlation time distribution, and the $(1-S^2)$ and S^2 dependence on order parameters is given by Eq. (3.19) and Eq. (4.9) of Brown (1982).

Page 4, line 106

We have added the citations, in addition to another citation (Beckmann, Phys. Rep. **1988**, **171(3)**, **85**), from which this particular notation partially originated.

Page 7, line 156 - please also cite Brown (1982) here.

Page 8, line 202.

We have added the citation.

Page 7, lines 169-172 - Please also cite the generalized two-step approach of Brown (1982). The references cited are limiting cases for isotropic orientational averaging and are not directly applicable to lipid membranes. The results in Refs. 21-25 are obtained as the limits of the generalized two-step approach for isotopically reorienting systems (proteins, micelles). The results in Brown (1979); Brown, Seelig, and Häberlen (1979); and Brown (1982) should not be overlooked.

Page 9, Line 222.

We have re-written this sentence to state that what we have achieved an explicit implementation of the theory of Brown and co-workers, and leave references also to Lipari/Szabo and Wennerström.

Page 10, line 246 - please consider citing Brown (1982) and Brown, Ribeiro, and Williams (1983) where are many of the concepts of collective lipid dynamics where first discussed in regard to NMR relaxation.

Page 12, line 308

We've added these citations.

Page 11 Fig. 5 - for the total motion, how is the axial symmetry about the director axis manifested in the energy landscape?

Edit: Page 18, line 440

We point out that symmetry around the director leads to tensors aligned with the chain (Figure 5C), and this leads to the uniform behavior in Figure 6D.

Page 12, Discussion - at the outset please re-define "detector" for the general reader.

Page 14, line 336

We've added the discussion.

Page 13, line 304 - here it is necessary to cite: Brown (1982) and Nevzorov, A. A., and Brown, M. F. (1997) J. Chem. Phys. 107, 10288-10310. See also Chakraborty, S., et al. (2020) Proc. Natl. Acad. Sci. U.S.A. 117, 21896-21905.

Edit: Page 15, Line 372

We've added these citations.

Page 20, line 542 - please cite Brown (1982) here for the case of lipid membranes.

Methods, Page 3, paragraph 2

This citation was misplaced. The citation for calculating the correlation function belongs two lines above, so we have moved it up and also added the citation to Brown (1982). On this line, we have replaced the citation with one to my own work (Smith, Angew. Chem. 2019) since we refer to a previous approach that we used for analyzing correlation functions.

Supplementary information:

Page 2 - I would recommend also citing Leftin, A., et al. (2014) Biophys. J. 107, 2274–2286 for assignments.

SI Page 2, line 40

We've added the citation

Page 5 - it is necessary to clearly define or re-define "detector" and "sensitivity" as these terms have different meanings for different readers.

SI Page 6, line 147-152

We define sensitivity more clearly here

SI Page 7, line 159-163

We define detector response more clearly here

Page 6 - I would recommend explicitly defining "detector" in words and referring to the equation in which it is explicitly defined.

See response to previous question

Page 18 - the authors should connect the coordinate frame analysis to the introduction of the director frame of the lipid bilayer. One possibility is to simply state that the director frame is taken as parallel to the lab frame thereby avoiding the orientation dependence. It is needed to clearly separate the time-dependent part of the interaction from the time independent part of the interaction. This is the most fundamental separation to be made, in my view.

This is a very interesting discussion point, and we will do our best to cover the critical aspects.

When we calculate the *total* correlation function for a given bond, we compute the orientationally averaged correlation function, and this is achieved by taking $C(t) = \frac{1}{2} \langle 3 \cos^2 \theta_{\tau, t+\tau} - 1 \rangle_{\tau}$, where $\theta_{\tau, t+\tau}$ is the angle between the tensor (i.e. bond) at some initial time (τ) and some later time ($t+\tau$). To get the average of this term, we never need an external reference frame, such as the director frame, since our reference frame is the tensor at time τ .

So what about the separation of the interaction into time independent and time dependent parts? If we have the director frame (or another symmetry frame), then we have one part of the tensor, with anisotropy $S\delta$ that is responsible for the coherent behavior of the spin system, and a second part of the tensor where $(1-S^2)\delta^2$ determines the relaxation behavior of the spin system. It certainly seems that we have *separated* the interaction, since S^2 from the coherent behavior and $(1-S^2)$ from the relaxation add to 1. But, if there is no symmetry axis, these two terms no longer sum to 1 (that is, we have to differentiate S as measured via residual couplings and $1-S^2$ that governs the relaxation, where the S are close to each other, but not identical).

Then, our view is that the separation into time-independent and time-dependent parts is not always possible in the sense that the contributions sum to yield the full tensor, and thus is not really fundamental (although it is extremely convenient if one has a symmetry axis). Keep in mind, that both S and $(1-S^2)$ are basically useful averages, for calculating particular behaviors of the spin system. But, are they fundamental? S may also not be the critical parameter for determining coherent behavior. Suppose we are at zero-field: then we may no longer take the high-field approximation and non-commuting components of the dipole coupling appear, thus a simple average of the dipole tensor will not sufficiently describe the coherent behavior.

Therefore, for the sake of generality, we choose not to take such a reference frame since the symmetry frame will not always exist for all system that we might treat with this method. We look forward to the reviewer's opinions on this point.

Page 21-33 - please make a stronger effort to connect the results of the mathematical analysis to physical observables in the case of lipid membranes. The impact of the work will benefit from showing how the mathematical formulation leads to new biophysical insights.

We think this is done more appropriately in the main text (see response to point 3), now better addressed with discussions of T_1 minima, S^2 vs. $1/T_1$ behavior, and the manifestation of power laws in the trends of detector responses.

Page 30, SI Figure 14 - the results for the overall motions seem to imply that a T_1 minimum might be observed under some combination of frequency and temperature. Could the authors please comment on this possibility?

As discussed in point 5 above, maxima in the total distribution of motion should result in T_1 minima when the maximum sensitivity of the T_1 experiment coincides with the maximum of the distribution of the total motion. Practically, this will not happen for most positions because the maximum of the distribution is occurring in the range of 1-10 ps, well away from the sensitivity of the T_1 experiment. Note further, that the separated motions, as shown in SI Fig 13, cannot be used to predict the T_1 minima—we must only consider the product of motions (SI Fig 15, main text Fig 6E), which often results in the maxima of the separated motions being covered up in the total motion.

We don't make further edits in the SI, but this is addressed in the main text (see point 5).

Page 32, SI Figure 16 - please comment on the cylindrical symmetry of the collective lipid motions about the director axis.
Main text Page 18, line 441

We agree that the manifestation of cylindrical symmetry is an aspect of the dynamics. We discuss that symmetry around the director leads to tensors aligning with their respective chains (Fig. 5C), leading to uniform behavior in Figure 6D. One should note, that the dynamic landscape captures amplitude and timescale of motion, but does not directly capture directionality of motion, so that symmetry may not so obviously manifest. This is why we have to refer to calculation of the residual tensors, where symmetry does manifest directly.

Michael F. Brown

Reviewer #2 (Remarks to the Author):

In the manuscript "A method to construct the dynamic landscape of a bio-membrane with experiment and simulation", the authors apply their recently developed "dynamics detectors" approach for analysis of conventional NMR relaxation and MD simulation data of fully hydrated bilayers of the previously thoroughly investigated phospholipid POPC, finally reporting their results as a "dynamic landscape" consistent with the extensive earlier literature on atomically resolved and multiple time scale studies of phospholipids. While all the individual aspects of the NMR experiments, MD simulations, "dynamics detectors" analysis, and "dynamic landscape" representation of the results can be found in the literature, the herein reported combined "method" can be taken as current state-of-the-art in investigations of molecular dynamics in lipid membranes and is of potentially great value for the broad community of biomembrane researchers. I recommend publication after only minor revisions as detailed below.

We thank the reviewer for the encouraging general remarks, and useful suggestions regarding presentation and context of the work.

1) The actual novelty of the manuscript is diluted by the far too liberal use of terms such as "novel", "we introduce", "we develop", "we obtain" for methodical aspects and molecular insights that are already well established in the literature. The text should be carefully edited throughout to give due credit to earlier literature and reserve the claims of novelty to the few – but significant – instances where it is warranted. More specifically:

We thank the reviewer for pointing out the excessive usage of such terms.

Page 1, line 14. Deleted 'novel' from "report a novel approach"

Page 8, line 202: Changed "introduce" to "apply"

Page 3, line 84 "develop a novel" changed to "apply a"

Page 18, line 437: deleted 'novel' from "apply a novel"

a) Abstract, "We develop a dynamics detector method...". The dynamics detectors method has already been described in a series of publications in Angew Chem and J Chem Phys: Refs 5, 15-17

Page 1, line 17

This is a good point: the detectors are previously developed. The novelty is in the development of the frame analysis and application of the detectors to the frame analysis. We've modified this: "We combine dynamics detector methodology with a new frame analysis of motion...".

b) Abstract, "...reveals vast differences in motion depending on molecular position.". These differences are well known in the literature on NMR relaxation and MD simulation studies of lipid membranes and corresponding synthetic model systems with surfactants. Ref 18 is a very recent summary of this area.

Page 1, line 21

Unfortunately, we can't add a reference in the abstract. However, we've replaced "reveals" with "shows", so that we aren't claiming to produce a result that is previously unknown. Other edits (see point d below) now give better credit especially to the work in ref. 18.

It is worth noting– ref. 18 shows a large difference in the effective correlation time especially in the backbone, so 'vast differences' certainly have been previously observed. However, our frame analysis furthermore indicates that backbone motion sees its largest contributions from slow, overall motion of the backbone as opposed to internal rearrangements (which are very small in the backbone), whereas internal rearrangement dominates motional amplitudes in the rest of the molecule. In this sense, we've revealed new differences not shown in ref. 18, by identifying even larger differences in the internal motion. Although, we do agree that changing the wording is reasonable.

c) Intro, "...we develop a novel frame analysis..." and "...we obtain a detailed characterization of the multidimensional dynamic landscape of a lipid membrane over several decades of correlation times"; page 9, "... we introduce a frame analysis..."; page 10, "... frame analysis provides unprecedented insights...". Cite and explain novelty of the current frame analysis in relation to the local frame decomposition and detailed characterization in Lindahl 2001. Molecular dynamics simulation of NMR relaxation rates and slow dynamics in lipid bilayers. J. Chem. Phys. 115, 4938-4950. doi: 10.1063/1.1389469

The concept of using a reference frame to remove some motion to better observe other motion is not really novel at all. What is unique in our study is that we apply the frames in such a way that we obtain the total correlation function from the product of correlation functions corresponding to several separated motions (thus being able to quantitatively evaluate

how individual motions contribute to the overall relaxation behavior/detector responses). Achieving this type of separation is non-trivial, because we need to calculate how motion of inner frames results in residual tensors and then carefully apply the motion of the outer frames to those residual tensors.

To the best of our knowledge, this was only achieved in such an explicit manner by Salvi, Abyzov, and Blackledge (Angew. Chem. Int. Ed, 2017, 56, 14020), and even in this case, their implementation was specific to protein backbone dynamics and only separated local librational motion, ϕ/ψ motion, and peptide plane tumbling, thus being primarily applicable to intrinsically disordered regions of proteins. Our approach is novel in its generality— one may use any system and any motion for which at least one frame can be defined (using any function of the atomic positions), and statistical independence can be achieved. Lindahl and Edholm use frames to separate one motion, but it is not separated in such a way to allow reconstruction of the total motion, and in any case, they use a rank-1 vector correlation function, which cannot be related to the relaxation directly (a rank-2 tensor correlation function is required).

Page 3, line 75: We have added the Lindahl and Edholm citation

Page 3, line 85: We clarify that the frame analysis yields correlation functions whose product yields the correlation function for the total motion (we also cite Salvi et al. here)

Page 8, line 207: We cite Salvi et al. for separation of the correlation functions for a special case.

d) Page 7. "...While obtaining full site resolution by combining MD and NMR is a step forward...". Clarify what is actually the step forward in comparison to the summary in ref 18 and the extensive original literature obtaining full site resolution by combining MD and NMR.

Page 7, line 181

We've re-phrased this to point out that previous studies have been able to obtain full site resolution of order parameters, or order parameters and the mean effective correlation time. The advancement in this study is that we furthermore obtain resolution in correlation time— we are able to look at amplitude of motion in different correlation time windows, giving us more information than is obtained just with the effective correlation time.

2) In abstract, replace the too general term "lipid membranes" with the more specific "fully hydrated POPC bilayers" to more clearly connect to previous literature on this in detail investigated model system.

Page 1, line 17

We've replaced "lipid membranes" with "fully hydrated POPC bilayers".

3) Intro. "...using an experimentally validated MD simulation, one should be able to extract and parameterize the specific motions." Cite the recent summary on this subject in Ref 18 already here.

Edit: Page 2, line 63

We've added the citation (Antila, J. Chem. Inf. Model, 2021).

4) Intro. To the list of previous spectroscopy and MD studies in Refs 10-14, add these ones focusing on <10-100 ns time scale motion of relevance for the current manuscript:

Lindahl 2001. Molecular dynamics simulation of NMR relaxation rates and slow dynamics in lipid bilayers. J. Chem. Phys. 115, 4938-4950. doi: 10.1063/1.1389469

Ferreira 2015. Model-free estimation of the effective correlation time for C-H bond reorientation in amphiphilic bilayers: 1H-13C solid-state NMR and MD simulations. J. Chem. Phys. 142, 044905. doi: 10.1063/1.4906274

Page 3, line 75

We've added these citations.

5) Page 10. "... the sign [...] of the residual tensor cannot be accessed with powder averaged samples". The sign can actually be determined with the S-DROSS method even for powder samples. For the method cite Gross 1997. Dipolar recoupling in MAS NMR: A probe for segmental order in lipid bilayers. J. Am. Chem. Soc. 119, 796-802. doi: 10.1021/ja962951b

and for the specific case of S-DROSS applied to POPC cite Ferreira 2016. Acyl chain disorder and azelaoyl orientation in lipid membranes containing oxidized lipids. Langmuir 32, 6524-6533. doi: 10.1021/acs.langmuir.6b00788

Page 12, line 280

Thank you for pointing this out to us. We've edited this to say that the sign can be determined with S-DROSS and cited both Gross and Ferreira.

6) Page 10. "...librational motion has very low amplitude...". Clarify that the librational motion is probably underestimated because of the LINCS constraint used in the simulation.

Thank you for pointing us to the LINCS algorithm, but unfortunately we have not been able to verify that this is true. LINCS constrains the bond lengths, and therefore has a large impact on vibrational motion, but only indirect influence on the librations. Specifically, Hess et al. (J. Comput. Chem. 1997, 18(12), 1463) point out that the LINCS constraint introduces a small rotation of the bond. Whether the angle of the rotation increases or decreases depends on if the LINCS algorithm decreases or increases the bond length to match the constrained value. Then, it is not clear to us that this would preferentially result in greater or smaller librations (in any case, Hess et al. say that the accuracy is "high enough for all purposes"). Of course, if there is a reference or stronger argument to the contrary, we'd be interested to understand this better and would be happy to make the change.

7) Page 10. "...attempting to estimate the specific form of $(1-S_2)\theta(z)$ is unlikely to return good results...". Explain that this is related to the non-uniqueness of the inverse Laplace transform, cite Istratov 999. Exponential analysis in physical phenomena. Rev. Sci. Instrum. 70, 1233-1257. doi: 10.1063/1.1149581

Page 12, line 298

We have added the discussion and citation.

8) Page 10. Cite Lindahl 2001 for previous power-law fits of $C(t)$ from MD of lipid membranes.

Page 12, line 308

We have added the citation.

9) Page 20. For inverse Laplace transform analysis of correlation functions, cite

Nowacka 2013. Signal intensities in 1H-13C CP and INEPT MAS NMR of liquid crystals. J. Magn. Reson. 230, 165-175.

doi: 10.1016/j.jmr.2013.02.016

Methods page 3 (page 25), para. 2, line 2-3

We've added the additional citation (and modified text slightly to fit better with the citation).

10) Eqs 1-3. Clarify that the detector analysis relies on the assumption that the rotational correlation function $C(t)$ is given by a sum of exponentials (Eqs 1 and 2 in ref 5). Also mention that the relation between $C(t)$ and $\theta(z)$ can be written as a Laplace transform.

Thank you for pointing this out.

Page 4, line 104

We have expanded the discussion on detector responses, and now show the equation for the correlation function, and say that we assume it is a sum of exponentials.

Page 4, line 108

We state that eq. 1 is a Laplace transform

11) Eq 3. Explain how the product assumption in Eq 3 is consistent with the sum of exponentials assumption underlying the detector analysis in Eqs 1-3.

Page 8, line 203

We point out that if the individual correlation functions are multi-exponential, so is their product (derivation in SI).

SI Page 17, line 349-364

We show that the product of two multi-exponential correlation functions is also multi-exponential, eq. (SI 7), and also provide the explicit form of the distribution of correlation times in terms of the distribution of correlation times for the original correlation functions, eq. (SI 8).

12) Figure 1. Give a tentative assignment of the microsecond dynamics captured by detectors 4 and 5 to some specific modes of motion (bilayer thickness fluctuations, lateral diffusion over curved bilayers,...). Also speculate over the underlying mechanism for the unexpected and previously not detected ~68 us mode of motion specific for carbon 9 at the double bond of the sn-2 acyl chain.

Thank you for the suggestions on what this motion may be.

Page 16, 386

The timescale is consistent with order fluctuations (local director motion) and diffusion, for the longer correlation times, so we state both these possible sources.

Page 16, line 390

We have added discussion about the larger detector response for carbon 9 in the range of ρ_5 . The behavior is rather surprising for us. Since DIPSHIFT indicates that the residual dipole coupling is very small at carbon 9, it is hard to see how a collective motion could bring about this behavior. A local, slow motion could be responsible, but given the timescale, we have not found it explicitly with the MD simulation. Another possibility is that there is some fluctuation of the chemical shift at carbon 9, yielding relaxation that we mis-interpret as reorientational dynamics.

13) Figure 4. Move molecules to avoid overlap with figure axes.

Page 11, line 264

We've updated the figure to avoid overlap

14) Figure 5. Point out that similar "dynamic landscapes" could also be obtained by regularized Laplace inversion of $C(t)$ as Fig 5 in Nowacka 2013.

Page 12, line 299.

We've added the reference and discussed the possibility of taking the inverse Laplace transform.

We think it's worth noting, however, that the inverse-Laplace transform is, even in the case of a simple motion, rather challenging to implement without adding some kind of unexpected behavior. For example, in a previous analysis, we used inverse Laplace transform followed by explicit calculation of detector responses (Smith et al, Angew. Chem. Int. Ed. 2019, 58, 9383). If you look in SI Fig. 8B of Smith et al. 2019, you'll note broad components at short correlation times, but narrower components at longer correlation times (I wonder if this is also happening in Fig 5a of Nowacka, noting that they have forced the distribution towards zero at the edges, yielding slightly different behavior). At least in our case, the change in breadth of the components is the result of varying timescale resolution. We can't distinguish correlation times of 1 ps

from those of 2 ps with a 5 ps timestep, and the same problem re-emerges at long correlation times, where you can't tell 5 μ s from 10 μ s if you only have 1 μ s of simulation. Then, these low-resolution regions end up with broader components, but this is only an artifact of the analysis, and is probably not a good representation of reality. Avoiding this problem is one of the motivations for detectors: we automatically obtain broad detectors where we have low timescale resolution and narrow detectors where we have high timescale resolution (this also works the same way for analyzing experimental data).

Reviewer #3 (Remarks to the Author):

In this work, the authors apply the concept of "dynamic detectors" to analyze motion in a lipid membrane. The idea of this analysis is to characterize the dynamics by constructing linear combinations of different NMR relaxation parameters that are expected to be specifically sensitive to different motional timescales. In this way, the detectors can in principle be used to identify the contribution of different time scales to the correlation function describing NMR relaxation. Although the authors claim it to be a "novel approach" in the abstract, it has been established for several applications in solid and liquid-state NMR. The novelty in this paper is therefore the application to the lipid membrane and the methods for separating out the contributions of different types of motion in the MD simulations to each detector (e.g. intramolecular vs overall motion etc.). Although the conclusions from that analysis are perhaps not unexpected, that may be because POPC lipid membranes are a very well-studied system, and application to other systems may prove more insightful.

Thank you for the kind remarks.

Reviewer #2 has also made a number of comments about novelty. It is correct to say that the detectors are not new, whereas the frame analysis to separate motion, as analyzed by detectors are new, and its application to lipids. We agree and have changed the manuscript in that regard. For the specific changes, we refer back to point one from reviewer 2, where we have made several edits to the manuscript.

Overall the paper is clearly written. Something that wasn't clear to me was how exactly the weights of the different observables in the detector functions are chosen. Is this done essentially by hand to locate a specific peak, or is there some systematic optimization done to target a desired profile/time window?

Thank you for pointing out that this remains unclear. It is discussed in the SI, but now we include it with its own heading, as section 1.5, and reference it in the main text (the details of detector optimization were somewhat hidden in a longer section about detector properties).

None of the detector optimization is done by hand anymore (in the initial paper, one used a MATLAB program and had to manually select positions in a multi-dimensional space to optimize detectors, see Smith et al. JCP 2018, doi.org/10.1063/1.5013316). In our current approach, detector optimization always begins with collecting the sensitivities of every experiment/data point into a matrix, and then performing singular value decomposition on that matrix. This allows us to reduce, say N data points into n fit parameters. For n fit parameters, these n singular values represent (approximately) the best possible fit of the data, which is equivalent to saying that we extract the maximum information from the experiments (or MD correlation function). We could increase n, and get more information, but also more noise. Returns are usually sharply diminishing with increasing n (see Smith et al. J. Biomol. NMR 2021 for application of SVD to determine information content of a data set, doi.org/10.1007/s10858-021-00361-1).

Once we have the n singular values, and corresponding vectors, we perform linear recombination of those vectors to obtain ideal detector sensitivities. If we want to compare two data sets (NMR v. MD, for example), we will use a linear least squares algorithm to approximately match the sensitivity of the MD detectors to the NMR detectors. On the other hand, when optimizing the NMR detectors, we just want them to be separated and narrow. To do this, we select some correlation time, and then optimize a linear combination of the vectors from SVD such that the sum (sensitivity) is equal to one at the given correlation time, and minimized but non-negative at all other correlation times (this is achieved with the linear programming algorithm). This yields narrow, but non-negative sensitivities. We then sweep through an array of correlation times, and what we find is that at most correlation times, the maximum of the sensitivity is greater than 1 and is located at a different correlation time than the value that we set to equal 1. However, at exactly n correlation times, the maximum is 1, and is located at the correlation time that we optimized to equal 1. These are the narrowest, most separated detector sensitivities. In this approach (which is used to optimize the initial, experimental detectors), the detector sensitivities are only determined by the choice of experiments (and the signal-to-noise of those experiments) and by the number of detectors used. Thus, they represent the true information content of the data set in a very systematic way.

Page 4, line 117

We point the reader to SI section 1.5

SI Page 9, line 220

We add a new heading here, to make it easier for the reader to find the section in the SI that describes how detector optimization is performed (previously combined with section 1.4)

REVIEWERS' COMMENTS

Reviewer #1 (Remarks to the Author):

The authors have revised their paper in response to the comments proved by me and the other reviewers which has led to substantial improvement and clarifications. Particularly the additional references to the literature have led to broadening the scientific impact of the work. I believe it is a valuable addition to the literature in this area and that publication in its present form is warranted.

General comments:

The authors refer to the absence of an average director frame in their analysis, and without going into all the details, I generally accept their statements. The usual relations using time-dependent perturbation theory within the so-called Redfield approximation are of course inapplicable to zero field or very slow motions. Consequently, the non-secular terms in the coupling Hamiltonian need to be considered, which as correctly pointed out are non-commuting with the main unperturbed coupling Hamiltonian. At a fundamental level the frame decompositions as implicit in the frame analysis are an application of the closure relation from group theory (see Refs. 26 and 41 of the revised text and see also Appendix of Molugu et al. Chem. Rev. 2017, 117, 12087). All the needed symmetry relations are included in the expansion over the frames, but of course disentangling the various detector responses or spectral densities is not trivial in closed form. However, they can be explored by MD simulations as in the present work. The authors clearly appreciate these aspects, and the approach is a useful way of exploring the problem.

To summarize the work captures the vitality of an important sub-field of membrane biophysics. It presents an alternative way of thinking about the broad correlation time distributions of lipid bilayers in relation to their molecular and material properties. Publication in its present form in Nature Communications is recommended.

Michael F. Brown